# On Learning Verifiers and Implications to Chain-of-Thought Reasoning

**Maria-Florina Balcan**
Carnegie Mellon University
ninamf@cs.cmu.edu

**Avrim Blum**
TTIC
avrim@ttic.edu

**Zhiyuan Li**
TTIC
zhiyuanli@ttic.edu

**Dravyansh Sharma**
TTIC, Northwestern University
dravy@ttic.edu

## Abstract

Chain-of-Thought reasoning has emerged as a powerful approach for solving complex mathematical and logical problems. However, it can often veer off track through incorrect or unsubstantiated inferences. Formal mathematical reasoning, which can be checked with a formal verifier, is one approach to addressing this issue. However, currently LLMs are simply not good enough to solve complex problems in a formal way, and even just formalizing an informal problem statement can be challenging. Motivated by this fact, in this work we consider the problem of learning reliable verifiers for sequential reasoning, including natural language Chain-of-Thought reasoning. That is, given a problem statement and step-by-step solution in natural language, the aim of the verifier is to output [Yes] if the reasoning steps in the solution are all valid, and [No] otherwise. In this work we give a formal PAC-learning framework for studying this problem. We propose and analyze several natural verification goals, at different levels of strength, in this framework. We provide sample complexity upper-bounds for learning verifiers satisfying these goals, as well as lower-bound and impossibility results for learning other natural verification objectives without additional assumptions.

## 1 Introduction

With increasing use of LLMs to solve complex mathematical and logical problems through chain-of-thought reasoning, it has become crucial to develop verifiers that can check the correctness of these generated solutions. In particular, even with recent advances, Chain-of-Thought (CoT) reasoning is still widely believed to suffer from catastrophic failures resulting from accumulated errors except for highly limited scenarios [LFL+23, SVK24]. It can be particularly challenging to detect subtle errors in long sequences of reasoning, especially when presented via informal natural expressions. This motivates the need for designing effective verifiers for CoT reasoning in natural language.

To study this problem, in this work we introduce a PAC-learning framework for learning verifiers for sequential reasoners. Our learning algorithms are given a sample of some problem statements and labeled reasoning sequences for the problems, and are required to check the correctness of unseen reasoning sequences for unseen problems. We consider several related but different verification goals and analyze the sample complexity for learning verifiers satisfying these criteria, giving both upper bounds and impossibility results.

For example, the simplest (weakest) verification goal we consider is that given a random reasoning trace from some underlying distribution $D$, the verifier should output whether the reasoning is correct

39th Conference on Neural Information Processing Systems (NeurIPS 2025).

or faulty (and if faulty, where the first error occurred), and it should have error rate at most some given $\epsilon > 0$. The aim is then, with probability $\geq 1 - \delta$, to learn such a verifier from labeled data of correct and faulty reasoning traces from the same distribution. One drawback of this simple verification goal is that it is not secure against adaptive use. For example, if an LLM reasoner is told by the verifier that a reasoning trace $x_0, x_1, ..., x_t$ is incorrect at the $i$th step, then a natural reaction is to back up and replace $x_i$ with some other step $x_i'$ and try again, and to keep trying until a new reasoning trace is found that succeeds. But there is now no guarantee the final trace produced is correct, both due to the multiple rounds of querying and because the new traces queried may now be out-of-distribution.

To address the above challenge, we also introduce a stronger, more trustworthy verification goal, in which given some distribution $D$ over *problem instances* $x_0$, for most $x_0 \sim D$ the verifier should not accept *any* faulty reasoning trace from $x_0$. Of course, such a verifier should also accept at least some *correct* reasoning traces from $x_0$, and we give upper and lower bounds depending on whether we allow the verifier to just accept a designated *gold standard* reasoning trace $g(x_0)$ or whether we require it accept a large fraction of all correct reasoning traces from $x_0$ without any additional assumptions. These verifiers are more robust to any distribution shift in the reasoning traces compared to what was available in the training set.

Overall, our work introduces a principled framework for designing verifiers for CoT reasoning using machine learning. Our learnability results highlight the usefulness of our framework for designing verifiers with desirable properties with bounded sample complexity and some fundamental requirements for learning CoT verifiers.

## 1.1 Contributions

- We introduce a formal framework for studying verifiers for Chain-of-Thought reasoning. Given any problem statement and a sequence of reasoning steps for the problem, we propose the problem of learning verifiers that examine the steps for correctness, and for an incorrect reasoning trace return the first faulty step in the reasoning.

- We formally define *simple verifiers* that have access to random Chain-of-Thought reasoning sequences labeled "correct" or "incorrect" along with the first faulty step. We establish sample complexity bounds for learning good simple verifiers in a PAC (probably approximately correct) sense for verifier classes that are finite or have a finite VC dimension.

- We next introduce the more powerful *trustable verifiers*, which only have access to random problems, and a *gold standard reasoner*, which provides a small number of guaranteed correct reasoning traces for each sampled problem. We establish PAC learnability of designing verifiers that accept all the gold standard reasoning traces on most problems and never accept faulty reasoning traces, provided that the space of reasoning steps is finite.

- We extend our trustable verification goal to the case where there may be a large number of gold standard reasoning traces, but only a random correct trace is available to the learner. We establish upper and lower bounds on the sample complexity of learning a verifier that is always sound (i.e., never accepts an incorrect trace) and accepts most of the gold standard traces on most problems.

## 1.2 Related work

*Chain-of-Thought generation.* Chain-of-Thought and its variants [WWS+22, ZZLS23, WWS+23, YYZ+23] are gaining popularity as paradigms for studying LLM reasoning. CoT reasoning has known connections with planning for complex tasks [HGM+23], and in fact, our theoretical abstraction applies to generating plans as well. [JVB+25] study the learnability of a time-invariant autoregressive generator for CoT for a fixed generation length $T$, and obtain sample complexity logarithmic in $T$, improving over the linear dependence for time-variant generation in [Mal24]. Their work focuses only on in-distribution generalization. In contrast, our *trustable* verification model is able to provide strong verification guarantees even for out-of-distribution reasoning, which is crucial in the context of typical CoT generation where the generator may adapt to prompts or feedback. Furthermore, we show concrete computational gaps between generation and verification, the functions in the generator class may not be efficiently evaluatable for a class of problems for which the verification functions are. We also note an equivalence between a special case of our verification model and their generation model, in the sense that an algorithm for one can be used to achieve the other (Remark 4.6). Empirically, LLM-based verifiers have been used to solve specific tasks, even outperforming

finetuning-based approaches [CKB$^+$21], especially with step-by-step verification [LKB$^+$24]. Our trustable verifiers involve checking proofs with respect to some gold standard reasoners, which may be reminiscent of imitation learning [NHB$^+$21, YSAN22].

*Learning with one-sided error.* Our strongest verification model requires the verifier to not accept any incorrect proof but possibly miss some legitimate proofs. The formulation bears resemblance to prior work on learnability under one-sided error [Nat87, Kiv95, BB05]. In particular, our learning algorithm is similar to the closure algorithm proposed in this literature. Other related lines of work that offer similar reliability guarantees using closure-based algorithms include selective classification [RS88, EYW10], robustly-reliable learning [BBHS22, BHPS23, BS24, BS25] and learning in the presence of strategic improvements [ABN$^+$25, SS25]. In addition, we consider learning from only positively labeled traces (Section 4.2). A related direction studies learning from positive and unlabeled data for binary classification [Den98, DGL05].

*Formal methods and learning.* Formal verification [CW96] is a sound approach used to verify the correctness of software or mathematical proofs written according to precise formal specifications. Although LLMs have helped improve some formal verification systems [CP24], it is not clear if formal verification can be used to verify the natural language reasoning of modern LLMs [ZSL$^+$24]. A related approach is to use proofs in a functional programming language (say Lean) to build datasets for training machine learning models capable of writing proofs in that formal language (while they also generate natural language descriptions of their proof steps, the evaluation is through correctness of the lean proof). Note that the "verifier" in this interaction is the Lean proof system, which only accepts formal language, while our work studies the learnability of verifiers that may evaluate natural language reasoning directly.

*Interactive provers.* Our work is related to a recent line of work related to interactive provers. Here, the goal is to generate a good prover that not only produces a good solution, but is also able to convince a given verifier $V$ about the correctness of its proof. This is complementary to our work in the sense that we show how to actually learn good verifiers from data, which can, in turn, provide good automated benchmarks with respect to which the prover establishes verifiability guarantees. [GRSY21] define a PAC model where the prover is able to convince the verifier on most typical inputs, analogous to our SVPAC model. [AGPR24] develop more powerful interactive provers that provide per-instance guarantees, similar to our TVPAC models. In this context, our work opens up concrete questions for interactive verification—whether it is possible to achieve a better sample complexity of learning verifiers, either assuming provers with certain guarantees or with techniques from active learning [BBL06]. [BLM$^+$25, RSS$^+$25] show theoretical as well as practical advantages of verifier-assisted constrained language generation.

## 2 Setup and Definitions

Let $X$ denote a domain of possible problem statements. For example, an $x_0 \in X$ could be a mathematical conjecture, or a Satisfiability problem instance, or the description of an initial state in a Sudoku game or Einstein puzzle. Let $\Sigma$ denote a set of possible reasoning steps; we will think of a "step" as a few tokens, such as [Suppose, for contradiction, that $\sqrt{2} = \frac{a}{b}$ for integers $a, b$] or [Clauses $(A \vee B)$ and $(A \vee \neg B)$ imply $(A)$]. A *verifier* is a function $h : X \times \Sigma^* \to \{\text{YES}, \text{NO}\}$, where given input $(x_0, \tau = (x_1, x_2, ..., x_t))$ where $x_0 \in X$ and each $x_i \in \Sigma$ for $i \geq 1$, the verifier should output YES if $x_t$ is a legitimate inference from $(x_0, (x_1, ..., x_{t-1}))$ and should output NO if $x_t$ is not a legitimate inference from $(x_0, (x_1, ..., x_{t-1}))$. Formally, we can allow $h$ to output arbitrarily if $(x_0, (x_1, ..., x_{t-1}))$ itself contains a faulty step: that is, a "correct" $h$ only needs to output correctly on $(x_0, (x_1, x_2, ..., x_t))$ if $(x_0, (x_1, ..., x_{t-1}))$ is itself correct.

Given a full reasoning trace or proof $(x_0, (x_1, ..., x_T))$, a verifier $h$ is "run" on the trace by running $h$ on each prefix, that is, $h(x_0, (x_1)), h(x_0, (x_1, x_2)), ..., h(x_0, (x_1, ..., x_T))$. If all of those runs output YES then we define $h$ as saying the reasoning is legitimate, and if any output NO then we define $h$ as saying the reasoning is faulty (and we output the first NO as the location of the first faulty step). We will use $H$ to denote a family of verifiers.

*Remark* 2.1. We will typically want to emulate the behavior of (or, in agnostic verification, at least be competitive with) the best verifier $h^* \in H$. In particular, note that our formulation does not involve an explicit notion of verifying whether the reasoning trace "completes" the proof. A good verifier may "accept" a reasoning trace if $h^*$ accepts it. This is analogous to the notion of CoT generation

| Verifier | SVPAC (Sec. 3) | TVPAC (Sec. 4.1) | $\gamma$-TVPAC (Sec. 4.2) |
|---|---|---|---|
| Data format | random tuples (problem, reasoning, first incorrect step) | (random problem, $\leq k$ gold-standard solutions) | random pairs (problem, correct reasoning) |
| Learning Algorithm | ERM on training set | ERM using trees $\mathcal{T}_g(x)$ | Intersection of all consistent verifiers (Algorithm 1) |
| Sample complexity (finite $H$) | $\tilde{O}(\log|H|)$ | $\tilde{O}(\log|H|)$ | $\tilde{\Theta}(|H|)$ |
| Sample complexity (bounded VCdim($H$)) | $\tilde{O}(\text{VCdim}(H))$ | $\tilde{O}(\text{VCdim}(H))$ | $\tilde{O}(\text{VCdim}(H))$ (if intersection-closed) |

Table 1: Different verification goals, training data, learning algorithms and sample complexities. The soft-O and soft-$\Theta$ notation suppresses dependence on quantities apart from $|H|$ and VCdim($H$).

studied by [JVB$^+$25], where one hopes to emulate the "correct" generator, although they do not consider the extension to agnostic learning (see Appendix F).

*Remark* 2.2. Our use of Chain-of-Thought is different from that in the original paper [WWS$^+$22], where it is used primarily as a prompting technique. We use CoT to refer to the standard generation pattern of reasoning models such as o3, Deepseek R1, that is, the model generates the ordered list of intermediate reasoning steps $(s_1, \ldots, s_T)$ before its final answer. Our use of the terminology Chain-of-Thought is consistent with prior theoretical work, e.g. [JVB$^+$25, Mal24] which studies Chain-of-Thought generation (in contrast, we study verification of the reasoning produced by such models). Suppose that the language model is given an input question and it produces an output consisting of intermediate steps before arriving at the final answer. This behavior may be due to the nature of the training data with such Chain-of-Thought sequences as studied in the prior works mentioned above, or because the language model was prompted to "think step by step".

## 3 Simple Verification

Let $D$ be a distribution over problems and reasoning traces $(x_0, (x_1, ..., x_t))$ of length $\leq T$, including both legitimate reasoning traces and faulty reasoning traces. Assume that we have an i.i.d. training sample $S$ of problems and reasoning traces drawn from $D$, and the traces are labeled according to a perfect verifier $h^* \in H \subseteq \{\text{YES}, \text{NO}\}^{X \times \Sigma^*}$. That is, a trace is labeled YES if every step in it is legitimate, and is labeled NO otherwise. Assume that for the faulty traces, we are also told which is the first faulty step in it. We aim to learn a verifier $h$ from such a sample that has a small error over unseen samples from $D$. Note that we make no assumptions on the size of $\Sigma$ (the set of all possible reasoning steps) for this result.

**Goal:** Given the training set $S$ of reasoning traces drawn i.i.d. from $D$, our goal is to learn a *simple verifier* $h$ with error at most $\epsilon$ over $D$. Specifically, given a new trace $(x_0, (x_1, \ldots, x_t)) \sim D$, we will run $h(x_0, (x_1)), h(x_0, (x_1, x_2)), \ldots, h(x_0, (x_1, \ldots, x_t))$ and if all of them output YES then we say the reasoning trace is "legitimate" and if any output NO then we say the reasoning is "faulty", and we output the first NO as the location of the first faulty step. We say that the learned verifier $h$ is correct on trace $(x_0, (x_1, \ldots, x_t))$ if either

(a) the entire trace consists of correct reasoning steps (i.e., $h^*(x_0, (x_1, \ldots, x_j)) = \text{YES}$ for all $1 \leq j \leq t$) and all of $h(x_0, (x_1)), h(x_0, (x_1, x_2)), ..., h(x_0, (x_1, \ldots, x_t))$ output YES, or

(b) the trace is faulty reasoning and $h$ correctly outputs NO on the first faulty step (and outputs YES up until the first faulty step).

Any other behavior is viewed as $h$ making an error on the given reasoning trace.

We will use $\mathsf{f}(h, (x_0, \tau = (x_1, x_2, ..., x_t)))$ to denote the smallest index $j$ such that the verifier $h$ rejects the reasoning sub-trace $(x_1, \ldots, x_j)$, that is $h(x_0, (x_1, \ldots, x_j)) = \text{NO}$, and set to $t$ otherwise (if no such index exists). That is, $\mathsf{f}(h, (x_0, \tau))$ is the index of the reasoning trace $\tau$ where $h$ terminates its evaluation of $(x_0, \tau)$, either by finding a faulty step at some index $j \in [t]$ or accepting the reasoning

as legitimate by evaluating to YES all the way through the last index $t$. We use this to define the following loss function which gives the 0-1 loss of the verifier $h$ on the input $(x_0, \tau)$

$$\ell_h(x_0, \tau) = \ell_{h^*}(h, (x_0, \tau)) := \mathbb{I}[h(x_0, \tau_j) \neq h^*(x_0, \tau_j) \text{ for some } j \leq \mathsf{f}(h^*, (x_0, \tau))].$$

Here $\tau_j = (x_1, \ldots, x_j)$ denotes a sub-trace of $\tau = (x_1, \ldots, x_t)$. Formally, we have the following definition for simply-verifiably-PAC learning a verifier from a class of verifiers $H$.

**Definition 3.1** (SVPAC-learnable). Let $X$ denote the problem space and let $H \subseteq \{\mathsf{YES}, \mathsf{NO}\}^{X \times \Sigma^*}$ denote the class of verifiers. Then a learner is said to simply-verifiably-PAC learn $H$ with sample size $m = M(\epsilon, \delta)$ (sample complexity is the smallest such $m$) if for any $h^* \in H$, for any $\epsilon, \delta \in (0, 1)$, for any distribution $D$ over $X \times \Sigma^*$ realizable by $h^*$ (i.e., legitimate inference is always given by $h^*$), given a sample $S \sim D^m$, the learner outputs a verifier $h$ such that with probability at least $1 - \delta$ over the draw of $S$, $\Pr_{(x_0, \tau=(x_1, \ldots, x_t)) \sim D}[\ell_{h^*}(h, (x_0, \tau)) = 1] \leq \epsilon$.

The learner is said to be proper if $h \in H$. Note that our definition above requires the learned verifier $h$ to match the behavior of the correct verifier $h^*$ (with high probability) on any new reasoning trace drawn from $D$ up to the first faulty step (if one exists) pointed out by $h^*$. We will now show that it is possible to learn such a verifier with a small sample complexity. First, for the case of finite class of verifiers $H$, we observe that a simple union-bound based argument implies that we can learn a good verifier with $O(\log |H|)$ trace samples. See Appendix B for a proof.

**Theorem 3.2.** Any finite class of verifiers $H$ is SVPAC-learnable with sample complexity $\frac{1}{\epsilon}(\log(|H|) + \log \frac{1}{\delta})$.

We further show that a finite VC dimension of the verifier class is a sufficient condition to SVPAC-learn with respect to $H$. Our sample complexity bounds in this case are $O(\mathsf{VCDim}(H) \log T)$, scaling only logarithmically with the maximum length $T$ of a reasoning trace. We will select $h \in H$ by ERM (Empirical Risk Minimization) over the training sample. Note that we will run a verifier $h$ up to $T$ times on any sample trace to determine whether it runs correctly on it. Our argument adapts the analogous proof in [JVB+25]. A complete proof is in Appendix B.

**Theorem 3.3.** Any class of verifiers $H$ with finite VC-dimension $\mathsf{VCDim}(H)$ is SVPAC-learnable with sample complexity $O\left(\frac{1}{\epsilon}(\mathsf{VCDim}(H) \log T + \log \frac{1}{\delta})\right)$.

Our model for the simple verifiers above allows for learning a verifier from an arbitrary unknown fixed distribution $D$ over the reasoning traces. However, a major limitation of this model is that the guarantees only apply to traces drawn according to $D$. If a reasoning model is told that there is a faulty step in its reasoning chain $(x_1, \ldots, x_n)$, then it might modify its reasoning slightly to $(x_1, \ldots, x_n')$. But the new trace is no longer from $D$ and a verifier trained over samples from $D$ is not guaranteed to work well on this modified reasoning trace. In other words, the feedback from the verifier may be the very reason why there is a distribution shift. In the following sections, we introduce a more powerful model for learning verifiers that are robust to distribution shifts that may be induced as a natural consequence of receiving feedback from the verifier.

## 4 Trustable Verification

As discussed above, designing a verifier that only works well for in-distribution reasoning traces may not be desirable in typical scenarios. Motivated by this, we introduce a model for learning more powerful verifiers which provide strong guarantees for *any reasoning trace*, as long as the problem statements come from a distribution. In particular, we require that for most problem statements, the learned verifiers do not accept *any* false traces; that is, the learner should be *sound*. However, we potentially relax the requirement that the learner must accept all correct traces. It turns out that we observe two distinct regimes for learnability depending on whether the number of correct reasoning traces is small (and are all available for training) or large (and only a random trace per problem is given in the training set).

**Assumptions.** We will make two additional assumptions in order to achieve the above stronger verification guarantee. First, we assume that correct proofs on any problem $x$ are given to the learner

by a *gold standard* reasoner $g : X \to 2^{\Sigma^T}$. That is, $g(x)$ denotes a set of correct reasoning traces for problem $x$, and we will have access to some reasoning traces (made more precise below) generated by $g$ in our training set. For example, $|g(x)| = 1$ corresponds to there being a single correct gold standard reasoning trace for the problem $x$, which will be available if the problem $x$ is sampled in the training set. A caveat is that we would not be able to verify reasoning traces that are not generated by the gold standard reasoner available to us, even if they may be legitimate. Second, we will assume that the set of legal reasoning steps $|\Sigma|$ is finite.

**Goal:** Our training set $S$ will consist of $m$ problems drawn i.i.d. from some distribution $D$. For each problem $x$ in the training set, we will run $g$ to create the gold-standard traces, which will be our positive examples. If the number of correct traces is small, we can create negative examples for each way of deviating from the tree of gold-standard proofs (see Section 4.1). Given these examples, our goal is to learn a *trustable verifier* $h$ that, given a new problem $x \sim D$ and a proposed reasoning trace $\tau$ for it, is able to verify (with high probability) if the reasoning trace is correct according to $g$. That is, $h$ is correct on $x$ if it will reject *all* faulty traces on $x$, and will correctly accept *most* (or even *all*) traces that match the gold standard $g$.

Using terminology familiar from formal logic, we define the goal for our learned verifiers in terms of sound and complete verification as stated below.

**Definition 4.1** ($\gamma$-completeness w.r.t. $g$ and $\tilde{D}_{|x}$; stepwise and overall soundness). Given a problem $x \in X$, a set of correct reasoning traces $g(x) \subseteq \Sigma^T$ for the problem, and a distribution $\tilde{D}_{|x}$ over traces in $g(x)$, a verifier $h : X \times \Sigma^T \to \{\text{YES}, \text{NO}\}$ is said to $\gamma$-completely verify $x$ w.r.t. $g$ and $\tilde{D}_{|x}$ if $C_h(x) = \{\tau \in \Sigma^T \mid h(x, \tau_j) = \text{YES} \ \forall \text{ prefixes } \tau_j \text{ of } \tau\}$ satisfies $\mathbb{E}_{\tilde{D}_{|x}}[C_h(x) \cap g(x)] \geq \gamma$. Furthermore, we say that $h$ 1-completely verifies $x$ w.r.t. $g$ if $g(x) \subseteq C_h(x)$.

$h$ is said to *stepwise* soundly verify $x$ if whenever $h(x, \tau_j) = \text{YES}$ for $\tau_j \in \Sigma^{\leq T}$, then $\tau_j$ is a prefix of some $\tau \in g(x)$ (in particular, this implies $C_h(x) \subseteq g(x)$), and is said to *overall* soundly verify $x$ if $C_h(x) \subseteq g(x)$.

1-completeness corresponds to the learner essentially accepting all the traces that the gold reasoner $g$ deems as correct. In other words, 1-completeness w.r.t. $g$ (omitting the conditional distribution $\tilde{D}_{|x}$) means that $\gamma$-completeness holds in the above definition for $\gamma = 1$ for all conditional distributions. Later, we will relax 1-completeness to $\gamma = 1 - \eta$ completeness for small $\eta$ in some more challenging learning settings. We study two types of soundness guarantees—a stronger "stepwise" soundness which guarantees that we reject a proof at the first incorrect step (deviation from a gold-standard step) and "overall" soundness that only guarantees that incorrect proofs of length $T$ are rejected. Note that stepwise sound verification implies overall sound verification, but the converse may not necessarily hold.

*Remark* 4.2. We note that soundness and completeness of proof systems is a terminology also used in formal verification and logic, and caution the reader against conflating them with our notions. A soundness guarantee for a deductive system expresses that all provable sentences are true. Completeness states that all true sentences are provable. While there is an analogy, we remind the reader that our study applies to natural language reasoning while formal logic involves proofs expressed in a very precise formal language and their verification.

## 4.1 Sample complexity when the number of correct proofs is small

In this section, we will assume that the number of gold standard reasoning traces for any problem of interest in $X$ is small. That is, $|g(x)|$ is bounded by a small constant $k$ for any $x \in X$[1]. In this case, it is reasonable to expect that we have access to all the gold standard proofs for any problem $x$ in the training sample. We show how to create training samples for learning a verifier using $g$ and establish sample complexity bounds for learning verifier classes that are finite or have finite VC dimension.

Formally, for each problem $x$ in the training sample $S \sim D^m$, we will run $g$ to generate all the gold standard proofs. These will be our positive examples. To generate negative examples, we consider the first step of deviation from any correct trace for $x$ and add a negative example corresponding to it. Let

---

[1] A natural example for the case $k = 1$ could be a SAT-solver or an Mixed Integer Program solver where the gold-standard solver $g$ uses a deterministic branching rule that we know works pretty well.

$\mathcal{T}_g(x)$ denote the tree of positive traces on the problem instance $x$. The root of the tree is the problem statement $x$, and each node represents a valid reasoning step according to one of the positive traces in $g(x)$. By assumption on $|g(x)|$, $\mathcal{T}_g(x)$ has at most $k$ leaf nodes. Now we create negative examples for each internal node $x_i$ of $\mathcal{T}_g(x)$ as follows. Let $(\tilde{x}_0 = x, \tilde{x}_1, \ldots, \tilde{x}_i = x_i)$ denote the path from the root to $x_i$ on $\mathcal{T}_g(x)$, and $X_i \subset \Sigma$ denote its set of child nodes. Then for every $x' \in \Sigma \setminus X_i$, we create a faulty trace $(\tilde{x}_0, \tilde{x}_1, \ldots, \tilde{x}_{i-1}, x')$ and add it as a negatively labeled example for the problem $x$.

Finally, we formally state the definition of *trustable verification*. Notably, we require the learned verifier to be both complete (w.r.t. the gold standard $g$) and sound on problems drawn from $D$. In contrast to simple verifiers, the traces that we expect a *trustable verifier* to verify can be arbitrary.

**Definition 4.3** (stepwise and overall TVPAC-learnable). Let $X$ denote the problem space and let $H \subseteq \{\text{YES}, \text{NO}\}^{X \times \Sigma^*}$ denote the class of verifiers. Let $g(x) \subseteq \Sigma^T$ denote the set of correct reasoning traces for any $x \in X$. Then a learner is said to stepwise (resp. overall) trustably-verifiably-PAC learn $H$ with sample size $m = M(\epsilon, \delta)$ (sample complexity is the smallest such $m$) if for any $h^* \in H$, for any $\epsilon, \delta \in (0, 1)$, for any distribution $D$ over $X$ realizable by $h^*$ (i.e. for all $x$, $g(x) = C_{h^*}(x) = \{\tau \in \Sigma^T \mid h^*(x, \tau_j) = \text{YES} \ \forall \text{ prefixes } \tau_j \text{ of } \tau\}$), given a sample $S \sim D^m$ and for each $x \in S$ given access to the set $g(x)$, the learner outputs a verifier $h$ such that with probability at least $1 - \delta$ over the draw of $S$, $\Pr_{x \sim D}[h \text{ is 1-complete w.r.t. } g \text{ and stepwise (resp. overall) sound for } x] \geq 1 - \epsilon$. The learner is said to be proper if $h \in H$.

For the case of a finite verifier class $H$, we can still show a $O(\log |H|)$ upper bound on the sample complexity of learning a good verifier. A proof is located in Appendix C.

**Theorem 4.4.** Any finite class of verifiers $H$ is stepwise TVPAC-learnable with sample complexity $\frac{1}{\epsilon}(\log(|H|) + \log \frac{1}{\delta})$.

We further show that it is possible to stepwise TVPAC-learn any verifier class with finite VC-dimension (complete proof in Appendix C).

**Theorem 4.5.** Any class of verifiers $H$ with finite VC-dimension $\mathsf{VCDim}(H)$ is stepwise TVPAC-learnable with sample complexity $O\left(\frac{1}{\epsilon}(\mathsf{VCDim}(H) \log(kT|\Sigma|) + \log \frac{1}{\delta})\right)$, where $k$ is a bound on the number of correct proofs generated by $g$.

Some remarks are in order. Our stepwise trustable verification model has an interesting property that good verifiers in our models for any problem $x$ not only guarantee correctness of the reasoning steps so far, but also prompt the reasoner away from possibly legitimate reasoning steps which may not however result in a solution for the problem $x$. This additional stronger property is not achieved by overall TVPAC verifiers. In fact, for the special case $|g(x)| = k = 1$, our verification model is equivalent to the Chain-of-Thought autoregressive generation model of [JVB+25]. This is surprising as verifying a proof is usually believed to be easier than generating it (although formally an open question, for instance determining whether $\mathsf{P} \neq \mathsf{NP}$), but the strong "guiding" abilities of our verifiers can be used for generation.

*Remark* 4.6. For $k = 1$, our stepwise trustable verification model is equivalent to the generation model of [JVB+25] provided $|\Sigma|$ is small, in the sense that an efficient algorithm for verification implies an efficient algorithm for generation, and vice versa. To see this, given a stepwise sound verifier $h$ that is guaranteed to accept only the single gold standard trace $g(x)$, we can generate the correct proof using $h$ as follows. Run $h(x, \tau_0)$ for each $\tau_0 \in \Sigma$ until one of them, say $x_1$, yields YES. Now run $h(x, (x_1, \tau_1))$ for each $\tau_1$ until acceptance, and so on. Doing this $T$ times generates a proof for $x$ that matches $g(x)$. Conversely, to verify if a generator is correct on a problem $x$, we can simply match its reasoning trace against $g(x)$. An interesting consequence of this is that we can hope to use a good verifier to train a good reasoner.

Thus, there is an equivalence between stepwise TVPAC verifiers and CoT generators (provided the size of the reasoning space $|\Sigma|$ is small). However, the equivalence breaks down for large $|\Sigma|$, and does not hold for overall TVPAC verifiers. In the following remark, we show a computational gap between proving and verification. Namely, for the same problem space, the proof generation functions may not be efficiently evaluatable even though the verifier functions can be evaluated in polynomial time.

*Remark* 4.7. *Computational gap between CoT provers and TVPAC verifiers.* We first provide a concrete example showing computational separation between CoT generators and overall TVPAC

---

**Algorithm 1** Intersection of Consistent Verifiers

---

**Require:** Set of positive problem-trace examples $S = \{(x^{(1)}, \tau^{(1)}), \ldots, (x^{(m)}, \tau^{(m)})\}$ where $x^{(i)} \overset{\text{i.i.d.}}{\sim} D, \tau^{(i)} \overset{\text{i.i.d.}}{\sim} \tilde{D}_{|x^{(i)}}$, verifier class $H$.

1: $H_S \leftarrow \{h \in H \mid h(x, \tau) = 1 \text{ for all } (x, \tau) \in S\}$.       *{Set of verifiers consistent with S}*
2: **return** $h' : (x, \tau) \mapsto \wedge_{h \in H_S} h(x, \tau)$.    *{predict YES only when every consistent h predicts YES}*

---

verifiers. Let $X$ be the set of all satisfiable USAT (Unique-SAT)[2] problems in $n$ variables and consisting of $m$ clauses. Suppose each reasoning step consists of a single variable assignment, i.e., $\Sigma = [n] \times \{0, 1\}$. Then it is easy to construct a verifier function $h$ that is efficiently computable and is both complete and overall sound. Given a problem-proof pair $(x, \tau)$, the verifier simply checks that no variable is given multiple inconsistent assignments, each variable appearing in $x$ has an assignment, and each clause in $x$ is satisfied. On the other hand, by the Valiant-Vazirani theorem [VV85], USAT has no polynomial-time randomized algorithm (assuming $\text{RP} \neq \text{NP}$). Therefore, there is no efficiently computable proof generator for all problems in $X$. The above argument can also be extended to stepwise TVPAC verifiers by using an exponential-size $\Sigma$. In the USAT example above, we could consider one-step proofs ($\Sigma = \{0, 1\}^{[n]}$) which set all the variables in a single step. In this case, the reduction in Remark 4.6 is no longer polynomial time since $|\Sigma| = 2^n$.

## 4.2 Linear sample complexity for any number of correct proofs

We will now consider an extension to our trustable model where we no longer assume a small bound on the number of gold standard traces for every problem $x \in X$. This would make it unreasonable to expect the gold standard reasoner $g$ to generate all proofs for a given problem instance $x$. Instead, we would only require it to generate a random correct proof. For an example, one could think of randomized solvers for constraint satisfaction problems. We will relax the goal of being perfectly complete w.r.t. $g$ (Definition 4.1) to being almost perfectly complete, while still requiring the verifier to be (overall) sound. In this subsection, we assume soundness refers to overall soundness throughout.

Our training set $S$ will consist of problem-trace pairs $(x, \tau)$ where $\tau$ is a random correct trace from $g(x)$. We learn from only positively labeled examples. Formally, we have the following definition.

**Definition 4.8** ($\gamma$-TVPAC-learnable). Let $X$ denote the problem space and let $H \subseteq \{\text{YES}, \text{NO}\}^{X \times \Sigma^*}$ denote the class of verifiers. Let $g(x) \subseteq \Sigma^T$ denote the set of correct reasoning traces for any $x \in X$. Then a learner is said to $\gamma$-trustably-verifiably-PAC learn $H$ with sample size $m = M(\epsilon, \delta)$ (sample complexity is the smallest such $m$) if for any $h^* \in H$, for any $\epsilon, \delta \in (0, 1)$, for any distribution $D$ over $X$ realizable by $h^*$, given a sample $S \sim D^m$ and for each $x^{(i)} \in S$ given access to *one random trace* $\tau_{x^{(i)}} \in \Sigma^T$ sampled according to $\tilde{D}_{|x^{(i)}}$ over $g(x^{(i)})$, the learner outputs a verifier $h$ such that with probability at least $1 - \delta$ over the draw of $S$ and the traces, $\Pr_{x \sim D}[h \text{ is } \gamma\text{-complete w.r.t. } g \text{ and } \tilde{D}_x \text{ and a sound verifier for } x] \geq 1 - \epsilon$.

An interesting special case is where $\tilde{D}_{|x}$ is the uniform distribution over $g(x)$ for all $x$. Here, $g$ would uniformly select one of its correct proofs when queried for generating the training set, and $\gamma$-completeness corresponds to accepting at least a $\gamma$ fraction of the correct proofs of $g$. For this more challenging setting, we first show the existence of an improper learner that achieves learnability in the case where the verifier class $H$ is finite. Our algorithm (Algorithm 1) outputs the intersection (agreement region) of all consistent verifiers with the training set. We show a bound on the sample complexity of Algorithm 1 which is linear in $|H|$.

**Theorem 4.9.** Let $\eta \in (0, 1)$. For any finite class of verifiers $H$, Algorithm 1 $(1 - \eta)$-TVPAC-learns $H$ with sample complexity $O\left(\frac{1}{\eta\epsilon}(|H| + \log\frac{1}{\delta})\right)$. Moreover, Algorithm 1 never accepts a faulty trace for any problem $x \in X$.

---

[2]Unique-SAT is a promise problem, the decision version of which asks whether a given Boolean formula, which is promised have either zero or exactly one satisfying assignment, has exactly one satisfying truth assignment. The search version of the problem further asks to determine the unique assignment.

*Proof. Overview.* Let $D^+$ denote the joint distribution over problem-trace pairs $(x, \tau)$ induced by the marginal distribution $D$ and the conditional distribution $\tilde{D}$ used to sample positive traces from $g(x)$. We will show that the expected error of the verifier learned using Algorithm 1 on a test pair $(x, \tau) \sim D^+$ is at most $O\left(\frac{|H| + \log \frac{1}{\delta}}{m}\right)$ with probability at least $1 - \delta$. We will further show that the errors are one-sided, i.e. we never accept a faulty trace for any problem $x$. Finally, using the law of total expectation, we show that this implies the stated bound on the sample complexity.

*Bound on generalization error.* We define the population error of $h \in \{\text{YES}, \text{NO}\}^{X \times \Sigma^*}$ (any verifier, not necessarily in $H$) on positive examples as $L_{D^+}(h) := \Pr_{(x, \tau) \sim D^+}[h(x, \tau) = \text{NO}]$. For each verifier $h_i \in H$, let $p_{h_i} = \Pr_{(x, \tau) \sim D^+}[h_i(x, \tau) = \text{NO} \text{ and } h^*(x, \tau) = \text{YES}]$ be the probability that $h_i$ incorrectly rejects a valid reasoning trace.

By the realizability assumption, $h^* \in H_S$ for any sample $S$ (recall that $H_S$ is the set of verifiers consistent with $S$, Algorithm 1). Since $h'(x, \tau) = \wedge_{h \in H_S} h(x, \tau)$, the error of $h'$ occurs only when at least one $h \in H_S$ incorrectly rejects a valid trace. Thus,

$$L_{D^+}(h') = \Pr_{(x, \tau) \sim D^+}[h'(x, \tau) = \text{NO} \text{ and } h^*(x, \tau) = \text{YES}]$$

$$= \Pr_{(x, \tau) \sim D^+}[\exists h \in H_S \text{ s.t. } h(x, \tau) = \text{NO} \text{ and } h^*(x, \tau) = \text{YES}] \quad \leq \sum_{h \in H_S} p_h.$$

For any subset $T \subseteq H$, define

$$L_{D^+}(T) := \Pr_{(x, \tau) \sim D^+}\left[\exists h \in T \text{ s.t. } h(x, \tau) = \text{NO} \text{ and } h^*(x, \tau) = \text{YES}\right].$$

Note that since $h'(x, \tau) = \bigwedge_{h \in H_S} h(x, \tau)$, we have the exact identity $L_{D^+}(h') = L_{D^+}(H_S)$.

Fix $\varepsilon > 0$ and let $\mathcal{T}_\varepsilon := \{T \subseteq H : L_{D^+}(T) \geq \varepsilon\}$. If $L_{D^+}(h') \geq \varepsilon$, then $H_S \in \mathcal{T}_\varepsilon$. Therefore, by union bound,

$$\Pr_S[L_{D^+}(h') \geq \varepsilon] = \Pr_S[H_S \in \mathcal{T}_\varepsilon] \leq \sum_{T \in \mathcal{T}_\varepsilon} \Pr_S[H_S = T] \leq \sum_{T \in \mathcal{T}_\varepsilon} \Pr_S[T \subseteq H_S].$$

For a fixed $T$, the event $T \subseteq H_S$ means that every $h \in T$ outputs YES on all $m$ samples. Equivalently, none of the $m$ samples falls into the event defining $L_{D^+}(T)$. Since the $m$ samples are i.i.d., we get

$$\Pr_S[T \subseteq H_S] = (1 - L_{D^+}(T))^m \leq (1 - \varepsilon)^m.$$

Thus,

$$\Pr_S[L_{D^+}(h') \geq \varepsilon] \leq |\mathcal{T}_\varepsilon| \cdot (1 - \varepsilon)^m \leq 2^{|H|} (1 - \varepsilon)^m \leq 2^{|H|} e^{-m\varepsilon}.$$

Setting the RHS to be at most $\delta$ gives $\varepsilon = \frac{|H| \ln 2 + \ln(1/\delta)}{m}$. Therefore, with probability at least $1 - \delta$,

$$L_{D^+}(h') \leq \frac{|H| \ln 2 + \ln(1/\delta)}{m}.$$

*We never accept a faulty trace.* By construction, $h'(x, \tau) = \wedge_{h \in H_S} h(x, \tau)$. This means $h'(x, \tau) = \text{YES}$ only if all $h \in H_S$ output YES for $(x, \tau)$. Since $H_S$ is set to be the set of all verifiers consistent with the training data $S$, and we assume by the realizability assumption that $h^* \in H$, we have $h^* \in H_S$. Therefore, if $h'(x, \tau) = \text{YES}$, then $h^*(x, \tau) = \text{YES}$ as well. This guarantees that $h'$ never accepts an invalid reasoning trace, i.e., $h'$ has zero false positive rate.

*Sample complexity bound.* We say that $x \in X$ is a *bad* problem if $h'$ is not $(1 - \eta)$-complete w.r.t. $g$ on $x$ (i.e., $h$ accepts fewer than $(1 - \eta)$ fraction of correct traces in $g(x)$ in expectation according to $\tilde{D}_{|x}$). We say that $\tau$ is a *bad* trace for a problem $x$, if $\tau$ is valid according to $g$ but not according to $h'$. If $h'$ makes an error on $(x, \tau)$, then either $x$ is a bad problem, or $x$ is not bad but $\tau$ is bad for $x$. Let $\epsilon = \Pr_D[x \text{ is bad}]$. The total error of $h'$, $L_{D^+}(h') \geq \epsilon \Pr_{\tilde{D}_{|x}}[\tau \text{ is bad} \mid x \text{ is bad}] \geq \epsilon \eta$. Using the above bound on $L_{D^+}(h')$, we get with probability $1 - \delta$, $\epsilon \eta \leq L_{D^+}(h') \leq \frac{|H| \ln 2 + \ln \frac{1}{\delta}}{m}$, which implies the claimed sample complexity bound. $\qquad \square$

Note that our upper bound above makes no assumption on $H$, other than it is finite. If $H$ is intersection-closed (that is, intersection of verifiers in $H$ is also in $H$), Algorithm 1 corresponds to the closure algorithm and $h' \in H$. In this case, we have much nicer bounds on the sample complexity—$\tilde{O}(\log |H|)$ for finite $H$ and $\tilde{O}(\mathsf{VCDim}(H))$ for $H$ with finite VC dimension (see Appendix D). As a simple example, suppose the set of reasoning steps $\Sigma$ consists of $n$ axioms. The verifier class $H$ consists of $2^n$ verifiers—corresponding to each subset $\sigma \subseteq \Sigma$, there is $h_\sigma \in H$ such that $h_\sigma$ only accepts traces that consist of reasoning steps from $\sigma$. In this case, the sample complexity of Algorithm 1 is $O(n)$ instead of $O(2^n)$. See Appendix E for additional examples.

**Lower Bounds.** We further show that the linear dependence on $|H|$ in our upper bounds on the sample complexity of trustable verification (given random access to positive proofs in the sense of Definition 4.8) is unavoidable without further assumptions on $H$. Roughly, if we do not have a bound on the number of correct reasoning traces from any given $x_0$, and if we want to learn a verifier $h \in H$ such that for most $x_0$, we have both (a) $h$ accepts *at least half* of the correct reasoning traces from $x_0$ and (b) $h$ rejects *all* faulty reasoning traces from $x_0$, then without further assumptions on which traces are correct, in the worst case we will need a training set with $\Omega(|H|)$ reasoning traces, for any $|H| \leq |\Sigma|^T$. This is in contrast to the $O(\log |H|)$ bound in Section 4.1 when we had only a single correct trace (or a few correct traces) per $x_0$.

Our first result states that if we want to output a sound proper verifier, i.e. $h \in H$ and we only require condition (b) above, then we already need at least $\Omega(|H|)$ samples to achieve TVPAC learnability for any learning algorithm. A proof is in Appendix C.

**Theorem 4.10.** Let $|\Sigma| \geq 2$. For each size $3 \leq \mathtt{H} \leq |\Sigma|^T$ there exists a finite class $H$ with $|H| = \mathtt{H}$ such that any proper learner that $\tilde{\epsilon}$-TVPAC learns $H$ (for any $\tilde{\epsilon} \geq 0$, i.e. the learned verifier is only required to be sound) has sample complexity at least $\Omega(|H|)$.

We next show that if we further require the learner to even accept at least a constant fraction of the correct traces (say $\frac{1}{2}$-completeness), in addition to soundness, then the linear lower bound on sample complexity holds even for representation independent learning, i.e. even if we allow the learner to output verifiers that are not in the verifier class $H$ (proof in Appendix C).

**Theorem 4.11.** Let $|\Sigma| \geq 2$. For each size $\mathtt{H} \leq |\Sigma|^T$ there exists a finite class $H$ with $|H| = \mathtt{H}$ such that any (proper or improper) learner that $\frac{1}{2}$-TVPAC learns $H$ has sample complexity at least $\Omega(|H|)$.

## 5  Discussion

Verification that can be trusted is a strong candidate approach towards powerful automated benchmarks for Chain-of-Thought reasoning. While verification using formal methods has been successfully deployed for testing software and proofs in formal systems, the task of verifying natural language reasoning seems more challenging. We propose a learning-based approach to designing such verifiers and introduce various verification models with different strengths of guarantees. Our simplest framework consists of verifiers that learn from random proofs from some fixed unknown distribution $D$ annotated with their first faulty step (or correct, if the entire proof is good). Such a verifier would be able to correctly annotate new reasoning sequences from the same distribution, but is not robust to distribution shifts (for example, due to adaptive editing of proofs by incorporating the feedback from the verifier). We next address a stronger type of verifiers that guarantee to reject *any* faulty reasoning (possibly very different from the incorrect proofs seen in the training set), by accepting only proofs that adhere to a certain *gold standard*. We call these verifiers *trustable* and show two distinct regimes for their learnability—small sample complexity when there is a small number of gold standard proofs for any problem, and an unavoidable larger sample complexity linear in the size of the verifier class without this assumption. This raises an interesting question—are there alternative assumptions or models for interactive verification where our linear lower bound on the sample complexity may be circumvented?

## Acknowledgments

We thank Feras Saad for pointing out an issue in the original proof of Theorem 4.9 which we subsequently fixed. This work was supported in part by the Simons Investigator Award MPS-SICS-00826333, a Microsoft Research Faculty Fellowship, and the National Science Foundation under grants CCF-2212968, ECCS-2216899, and ECCS-2216970.

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

# A  Additional Related Work

*Multiclass classification.* Our verifiers not only predict whether a proof is correct or faulty, but also indicate the first incorrect step in the chain of reasoning. The output of the classifier thus takes one of $T+1$ values (correct, or first fault at step $i \in [T]$) and can be thought of as a special type of structured multiclass classification. Multiclass classification has been extensively studied to understand how learnability is affected by the number of different label classes [Nat04, TB07], with a recent focus on infinite class size [BCD+22, HMR+23, HMZ24]. The latter raises an interesting open question regarding the learnability of CoT reasoners and verifiers for arbitrarily long traces.

*Exact learning.* Recent work [GLLS25] argues for the need to study exact learning [Ang88] for sound deductive reasoning. Our work represents advances towards exact learning for verifiers in several ways. First, our "trustable" verifiers are robust to distribution shifts of reasoning traces (although not of problem statements, so not exact learning)—in particular when we are "correct" for a problem statement, we are sound on all the reasoning traces for it. Second, we give an example where we achieve online verification with finite mistake bounds (see Example E.5) which can be viewed as an example of exact learning. Here we bound the total number of mistakes over problem and reasoning pairs on an arbitrary sequence. [GLLS25] argue that powerful verifiers may be useful in achieving exact learning for the reasoners as well.

# B  Omitted proofs from Section 3

We include here proof details for the results on SVPAC learning in Section 3.

## B.1  Proof of Theorem 3.2

*Proof.* We will simply output any verifier $h \in H$ that is consistent with the training sample (i.e. makes no error) and show that it achieves the desired low error for any sample size that is larger than the stated sample complexity. Fix some verifier $h$ with error $\geq \epsilon$ over $D$. This means that for a random reasoning trace $\mathbf{x} = (x_0, (x_1, ..., x_t)) \sim D$, with probability $\geq \epsilon$, $h$ makes a mistake, that is, $\ell_h(\mathbf{x}) = 1$. So, this means that the probability that $h$ does *not* make a mistake on any example $\mathbf{x} \in S$ is at most $(1-\epsilon)^{|S|}$. We now set this to $\delta/|H|$ and solve for $|S| = \frac{1}{\epsilon}(\log(|H|) + \log\frac{1}{\delta})$. $\qquad\square$

## B.2  Proof of Theorem 3.3

*Proof.* We will select $h \in H$ by ERM (Empirical Risk Minimization) over the training sample (in the realizable case this corresponds to selecting a consistent verifier). Note that we will run a verifier $h$ up to $T$ times on any sample trace to determine whether it runs correctly on it. We adapt the analogous proof for CoT generation due to [JVB+25]. Let $\tau_j$ be a shorthand for a reasoning sub-trace $(x_1, ..., x_j)$. Recall that the loss function on a given input $(x_0, \tau = (x_1, x_2, ..., x_t))$ is given as

$$\ell_h(x_0, \tau) = \mathbb{I}[h(x_0, \tau_j) \neq h^*(x_0, \tau_j) \text{ for some } j \leq \mathsf{f}(h^*, (x_0, \tau))],$$

and we define the corresponding function class $\mathcal{L}_{\mathcal{H}} = \{\ell_h \mid h \in H\}$.

Now, given a sample $S = ((x_0^{(1)}, \tau^{(1)}), \dots, (x_0^{(m)}, \tau^{(m)}))$ of size $m$, we are interested in the number of different behaviors of functions $h \in H$ over the sample. The shattering coefficient

$$
\begin{aligned}
\Gamma_{\mathcal{L}_{\mathcal{H}}}(S) &= |\{(\ell_h(x_0^{(1)}, \tau^{(1)}), \dots, \ell_h(x_0^{(m)}, \tau^{(m)})) \mid h \in H\}| \\
&\leq |\{(h(x_0^{(i)}, \tau_j^{(i)}))_{i\in[m], j\in[T]} \mid h \in H\}| \\
&\leq \Gamma_H(mT),
\end{aligned}
$$

where we have used that if $\ell_{h_1}(x_0, \tau) \neq \ell_{h_2}(x_0, \tau)$ then $h_1(x_0, \tau_j) \neq h_2(x_0, \tau_j)$ for some $j \in [T]$.

Using Sauer's lemma, for any $m \geq \frac{\mathsf{VCDim}(H)}{T}$, we have

$$\Gamma_{\mathcal{L}_{\mathcal{H}}}(m) \leq \Gamma_H(mT) \leq \left(\frac{emT}{\mathsf{VCDim}(H)}\right)^{\mathsf{VCDim}(H)}.$$

A standard lemma (e.g. [AB99], Appendix 1) now implies that $\mathsf{VCDim}(\mathcal{L}_\mathcal{H}) \leq \mathsf{VCDim}(H) \log T$, where $T$ is the maximum length of a reasoning trace. $\qquad\square$

## C   Proofs from Section 4

We include here proof details for our results on TVPAC learning in Section 4.

### C.1   Proof of Theorem 4.4

*Proof.* We will simply output any verifier $H$ that makes no error on the training sample. Assume that $h$ has error $\geq \epsilon$ over $D$. This means that for each $x_0 \in S$, with probability $\geq \epsilon$, $h$ will make a mistake on at least one of the examples created from $x_0$. Recall that a mistake here may be one of two kinds, either (a) $h$ accepts any other reasoning trace $f(x_0) \notin g(x_0)$, in which case $h$ must say YES to at least one of the negative examples in $S$ that was produced from $x_0$; specifically, it must have mistakenly accepted one of the traces $(x_0, ..., x_{i-1}, x_i')$ where $i$ is the index of the first step where $f(x_0)$ deviates from $\mathcal{T}_g(x_0)$, or (b) $h$ fails to accept some reasoning trace in $g(x_0)$, which is labeled YES in the sample. So, the probability that $h$ does *not* make a mistake on any example $x_0 \in S$ is at most $(1 - \epsilon)^{|S|}$. We now set this to $\delta/|H|$ and solve for $|S|$. $\qquad\square$

### C.2   Proof of Theorem 4.5

*Proof.* We select $h \in H$ by Empirical Risk Minimization over the augmented training sample (with positive and negative examples created using $g(x)$) described above (by realizability this corresponds to returning any consistent verifier). Note that we will run a verifier $h$ up to $kT|\Sigma|$ times on any sample trace to determine whether it runs correctly on it. The proof is similar to that of Theorem 3.3. Let $\tau_j$ be a shorthand for a reasoning sub-trace $(x_1, ..., x_j)$. Define a loss function on a given input $(x_0, \tau = (x_1, x_2, ..., x_t))$ as

$$\tilde{\ell}_h(x_0, \tau) := \mathbb{I}[h(x_0, \tau_j) \neq h^*(x_0, \tau_j)] \text{ for some } j \in [t],$$

where $h^*$ is the verifier in $H$ that accepts exactly the correct traces according to $g$, and let the corresponding function class be $\mathcal{L} = \{\tilde{\ell}_h \mid h \in H\}$.

Now given a sample $S = ((x_0^{(1)}, g(x_0^{(1)})), \dots, (x_0^{(m)}, g(x_0^{(m)})))$ of size $m$, we are interested in the number of different behaviors of functions $h \in H$ over the sample. Given a collection of correct traces $g(x_0)$, define $\tau_g^1(x_0)$ as the collection of all the sub-traces of traces in $g(x_0)$ along with one-step deviations of these sub-traces. Notice $|\tau_g^1(x_0)| \leq kT|\Sigma|$ for any $x_0$. The shattering coefficient

$$\begin{aligned}
\Gamma_\mathcal{L}(S) &= |\{(\tilde{\ell}_h(x_0^{(1)}, g(x_0^{(1)})), \dots, \tilde{\ell}_h(x_0^{(m)}, g(x_0^{(m)}))) \mid h \in H\}| \\
&\leq |\{(h(x_0^{(i)}, \tilde{\tau}))_{i \in [m], \tilde{\tau} \in \tau_g^1(x_0^{(i)})} \mid h \in H\}| \\
&\leq \Gamma_H(mkT|\Sigma|),
\end{aligned}$$

where we have used that if $\tilde{\ell}_{h_1}(x_0, \tau) \neq \tilde{\ell}_{h_2}(x_0, \tau)$ then $h_1(x_0, \tilde{\tau}) \neq h_2(x_0, \tilde{\tau})$ for some $\tilde{\tau} \in \tau_g^1(x_0)$.

Using Sauer's lemma, for any $m \geq \frac{\mathsf{VCDim}(H)}{kT|\Sigma|}$, we have

$$\Gamma_\mathcal{L}(m) \leq \Gamma_H(mkT|\Sigma|) \leq \left( \frac{emkT|\Sigma|}{\mathsf{VCDim}(H)} \right)^{\mathsf{VCDim}(H)}.$$

A standard lemma (e.g. [AB99], Appendix 1) now implies that $\mathsf{VCDim}(\mathcal{L}) \leq \mathsf{VCDim}(H) \log(kT|\Sigma|)$, where $T$ is the maximum length of a reasoning trace. $\qquad\square$

### C.3   Proof of Theorem 4.10

*Proof.* Select an arbitrary problem $x_0 \in X$ and set $D$ to be the constant distribution with support $\{x_0\}$. Also set the conditional trace generating distribution $\tilde{D}_{|x_0}$ to be the uniform distribution over $g(x_0)$ (we will set $g$ later). Let $|\Sigma| = b \geq 2$, so there are $b^T$ possible reasoning traces of length $T$

from $x_0$. Given $\mathtt{H} \leq b^T$, arbitrarily partition the $b^T$ reasoning traces into $\mathtt{H}$ disjoint sets $S_1, ..., S_{\mathtt{H}}$, each of size at least $\lfloor \frac{b^T}{\mathtt{H}} \rfloor$. Now, define the verifier class $H = \{h_1, ..., h_{\mathtt{H}}\}$ where $h_i$ accepts all reasoning traces *except* those in $S_i$. That is, if $C_h = \{t \in \Sigma^T \mid h(t) = \text{YES}\}$ denotes the set of traces accepted by $h$, then $C_{h_i} = \Sigma^T \setminus S_i$. Since we have no assumptions on which or how many traces are correct besides realizability, we stipulate that all $b^T$ traces are correct *except* for those in $S_{i^*}$ for some uniformly randomly chosen index $i^*$.

Now, a proper learner must output some $h_i \in H$. Suppose that the size of the training set $S$ is at most $\mathtt{H}/2$. The learning algorithm which is required to output some $h_i \in H$ can correctly choose $h_i = h_{i^*}$ with probability at most $2/\mathtt{H}$ since it is equally likely that any of the consistent verifiers is the right one. Note that in our construction $h_{i^*}$ is the only sound verifier in $H$. Thus, $\Pr[h \text{ is not sound}] \geq 1 - \frac{2}{\mathtt{H}} \geq 1 - \frac{2}{3} = \frac{1}{3}$. Thus, it is impossible to achieve error $\epsilon < \frac{1}{3}$ using $m \leq \mathtt{H}/2$ samples, establishing the desired lower bound of $\Omega(\mathtt{H})$. $\square$

### C.4 Proof of Theorem 4.11

*Proof.* Our initial setup is similar to the proof of Theorem 4.10. That is, we have the same $X = \{x_0\}, D, \tilde{D}_{|x_0}, g$ and $H$. For simplicity, assume that $\mathtt{H}$ is a multiple of 4.

Suppose the training set $S$ has size at most $\mathtt{H}/4$ (i.e. there are at most $\mathtt{H}/4$ labeled reasoning traces available, selected uniformly at random from $g(x_0)$). Any learned verifier $h$ that is $\frac{1}{2}$-complete (i.e. accepts at least half of the reasoning traces accepted by $h_{i^*}$) must accept traces from at least $\mathtt{H}/4$ distinct sets $S_i$ that were not observed in training data. Notice that these $\mathtt{H}/4$ sets constitute at least $1/3$ of the $3\mathtt{H}/4$ sets $S_i$ not observed in the training traces. This means that for $i^*$ randomly selected from these $3\mathtt{H}/4$ values, with probability at least $1/3$, $h$ accepts a trace in $S_{i^*}$. Thus any $\frac{1}{2}$-complete verifier fails to be sound with probability at least $\frac{1}{3}$. Thus, it is impossible to achieve error $\epsilon < \frac{1}{3}$ using $m \leq \mathtt{H}/4$ samples, establishing the desired lower bound of $\Omega(\mathtt{H})$. $\square$

## D   Intersection-closed Verifier Classes and $\gamma$-TVPAC Learning

The learnability of intersection-closed concept classes in the standard PAC model is a well-studied problem [HSW90, ACB98, AO07, Dar15]. Optimal sample complexity for these classes was known before Hanneke established the celebrated optimal bounds for (improper) PAC learning of arbitrary concept classes [Han16]. Here we will show that our lower bounds on sample complexity of arbitrary $\gamma$-TVPAC learning in Section 4.2 can be circumvented for intersection-closed verifier classes $H$. We will use $\mathcal{X} := X \times \Sigma^*$ to denote the domain of the verifiers. We start with some standard definitions restated in the context of verifier classes.

**Definition D.1** (Closure operator of a set). For any set $S \subseteq \mathcal{X}$ and any verifier class $H \subseteq 2^{\mathcal{X}}$, the *closure of $S$ with respect to $H$*, denoted by $\text{Clos}_H(S) : 2^{\mathcal{X}} \to 2^{\mathcal{X}}$, is defined as the intersection of all verifiers in $H$ that contain $S$, that is, $\text{Clos}_H(S) = \bigcap_{h \in H, S \subseteq h} h$.

In other words, the closure of $S$ is the smallest verifier in $H$ which contains $S$. If $\{h \in H : S \subseteq h\} = \emptyset$, then $\text{Clos}_H(S) = \mathcal{X}$. This allows us to formally define intersection-closed verifier classes.

**Definition D.2** (Intersection-closed classes). A verifier class $H \subset 2^{\mathcal{X}}$ is *intersection-closed* if for all finite $S \subseteq \mathcal{X}$, $\text{Clos}_H(S) \in H$. That is, the intersection of all verifiers in $H$ containing an arbitrary subset of the domain belongs to $H$. For finite verifier classes, this is equivalent to saying that for any $h_1, h_2 \in H$, the intersection $h_1 \cap h_2$ is also in $H$ [Nat87].

Examples of intersection-closed classes include axis-parallel $d$-dimensional hyperrectangles, intersections of halfspaces, $k$-CNF boolean functions, and subspaces of a linear space.

The *Closure algorithm* is a learning algorithm that generates a verifier by taking the closure of the positive examples in a given dataset, and negative examples do not influence the generated verifier (in fact, negative examples are not available in our $\gamma$-TVPAC model). The verifier returned by this algorithm is always the smallest verifier consistent with all of the positive examples seen so far in the training set. Note that Algorithm 1 is exactly the closure algorithm for intersection-closed verifier classes.

**Definition D.3** (Closure algorithm [Nat87, HSW90]). Let $S = \{(x_1, y_1 = f^*(x_1)), \ldots, (x_m, y_m = f^*(x_m))\}$ be a set of labeled examples, where $f^* \in H$, $x_i \in \mathcal{X}$ and $y_i \in \{0, 1\}$. The verifier $h_S^c$ produced by the closure algorithm is defined as:

$$h_S^c(x) = \begin{cases} 1, & \text{if } x \in \text{Clos}_H(\{x_i \in S : y_i = 1\}), \\ 0, & \text{otherwise.} \end{cases}$$

Here, $\text{Clos}_H(\{x_i \in S : y_i = 1\})$ denotes the closure of the set of positive examples in $S$ with respect to $H$.

The closure algorithm learns intersection-closed classes with VC dimension $d$ with an optimal sample complexity of $\Theta\left(\frac{1}{\epsilon}(d + \log \frac{1}{\delta})\right)$ [AO07, Dar15]. We can use this to establish $\gamma$-TVPAC learning for arbitrary intersection-closed verifier classes with a finite VC dimension. Note that our sample complexity bounds in this case are independent of the length $T$ of the reasoning trace.

**Theorem D.4.** Let $\eta \in (0, 1)$. Let $H$ be a class of verifiers that is intersection-closed and has a finite VC dimension $\mathsf{VCDim}(H)$. Algorithm 1 $(1 - \eta)$-TVPAC-learns $H$ with sample complexity $O\left(\frac{1}{\eta\epsilon}(\mathsf{VCDim}(H) + \log \frac{1}{\delta})\right)$. Moreover, Algorithm 1 never accepts a faulty trace for any problem $x \in X$.

*Proof.* Let $D^+$ denote the joint distribution over problem-trace pairs $(x, \tau)$ induced by the marginal distribution $D$ and the conditional distribution $\tilde{D}$ used to sample positive traces from $g(x)$. Note that in Algorithm 1 the intersection of consistent verifiers $h' \in H$ since $H$ is intersection-closed. We define the population error of $h \in H$ on positive examples as $L_{D^+}(h) := \Pr_{(x,\tau)\sim D^+}[h(x, \tau) = \text{NO}]$. Let $p_{h'} = \Pr_{(x,\tau)\sim D^+}[h'(x, \tau) = \text{NO} \text{ and } h^*(x, \tau) = \text{YES}]$ be the probability that $h'$ incorrectly rejects a valid reasoning trace.

By construction, $h'(x, \tau) = \text{YES}$ only if all consistent $h \in H_S$ output YES for $(x, \tau)$. Since we assume by the realizability assumption that $h^* \in H$, we have $h^* \in H_S$ which is the set of all verifiers consistent with the sample $S$. Therefore, if $h'(x, \tau) = \text{YES}$, then $h^*(x, \tau) = \text{YES}$ as well. Or, $h'$ never accepts an invalid reasoning trace.

Thus, $L_D(h') = L_{D^+}(h') = p_{h'}$. But, by known results for PAC learning of intersection-closed classes [AO07, Dar15], $m = O\left(\frac{1}{\varepsilon}(\mathsf{VCDim}(H) + \log \frac{1}{\delta})\right)$ training examples are sufficient to ensure $L_{D^+}(h') \leq \varepsilon$. As argued in the proof of Theorem 4.9, we have $\eta\epsilon \leq L_{D^+}(h')$, which establishes the claimed sample complexity. $\qquad\square$

We have the following corollary for learning finite and intersection-closed verifier classes $H$.

**Corollary D.5.** For finite intersection-closed $H$, Algorithm 1 $(1 - \eta)$-TVPAC-learns $H$ with sample complexity $O\left(\frac{1}{\eta\epsilon}(\log(|H|) + \log \frac{1}{\delta})\right)$.

## E    Examples

Here we will see several examples to illustrate our verification model. We start with a simple interval-based toy example which shows that SVPAC and $\gamma$-TVPAC learning may be possible even when $H$ and $\Sigma$ are infinite.

**Example E.1** (A toy example with interval verifiers). Let $X = \Sigma = \mathbb{R}$. The verifier class consists of functions

$$H = \{h_{r_1, r_2} : (x_0, \tau = (x_1, \ldots, x_i)) \mapsto \mathbb{I}[r_1 \leq x_0 - \sum_{j=1}^{i} x_j \leq r_2] \mid r_1, r_2 \in \mathbb{R}_{\geq 0}, r_1 \leq r_2\}.$$

That is, all reasoning traces for which the sum of reasoning steps is at some distance from $x_0$ that is within an unknown interval $[r_1, r_2]$ are valid. Notably, both $\Sigma$ and $H$ are infinite here. But $\mathsf{VCDim}(H) \leq 2$. For example, the training set consisting of the following reasoning traces

$$S = \{(0, (1)), (1, (3)), (2, (2, 3))\}$$

cannot be labeled $\{\text{YES}, \text{NO}, \text{YES}\}$ by any $h \in H$. This is because the distance of the trace sum from the problem $x_0 - \sum_{j=1}^{i} x_j$ for the training points are $1, 2$, and $3$ respectively. So, any $h_{r_1, r_2}$

which labels $(0, (1))$ and $(2, (2,3))$ as YES must also label $(1, (3))$ as YES. The finite VC dimension bound implies $H$ is SVPAC learnable with sample complexity $O\left(\frac{1}{\epsilon} \log \frac{1}{\delta}\right)$ by Theorem 3.3. Our results in Section 4.1 for 1-complete and sound verification do not apply as $|\Sigma|$ is not finite, but interestingly, the verifier class is still $\gamma$-TVPAC learnable (by Theorem D.4) with sample complexity $O\left(\frac{1}{\epsilon} \log \frac{1}{\delta}\right)$ since $H$ is intersection-closed.

The following example is a simple extension of the *autoregressive linear thresholds* studied as a family of Chain-of-Thought generators by [JVB$^+$25]. Intuitively, for token space $\Sigma = \{0,1\}$, a linear threshold $w \in \mathbb{R}^d$ looks at the last $l = \min\{|x|, d-1\}$ bits of the text $x$ generated so far and generates the next bit as $\mathbb{I}[w_1 + w[-l :]x[-l :]] \geq 0$, where $a[-l :]$ denotes the last $l$ elements (coordinates or tokens) of $a$. Instead, here we use linear thresholds for verification of reasoning traces as described below. In this case, the binary classes induced by the linear thresholds more naturally correspond to the outcomes {YES, NO} of verification (while generation beyond binary tokens needs some extension).

**Example E.2** (Linear threshold verifiers). Let $X = \mathbb{R}$, $\Sigma \subset \mathbb{R}$, $|\Sigma| = s$. The verifier class consists of functions induced by $d$-dimensional linear thresholds

$$H = \{h_{w,w_0} : (x_0, \tau) \mapsto \mathbb{I}[w_0 + w_1 x_0 + w[-l :]\tau[-l :] \geq 0] \mid w \in \mathbb{R}^d, w_0 \in \mathbb{R}, l = \min\{|\tau|, d-1\}\}.$$

Thus on a given problem and reasoning trace $(x_0, \tau)$, the verifier applies a linear threshold to the problem $x_0$ and the last $d-1$ reasoning steps (or all reasoning steps if $|\tau| \leq d-1$). Note that $H$ is SVPAC learnable with sample complexity $O\left(\frac{1}{\epsilon}(d + \log \frac{1}{\delta})\right)$ by Theorem 3.3. Similarly, we get a sample complexity of $O\left(\frac{1}{\epsilon}(d \log(ksT) + \log \frac{1}{\delta})\right)$ for TVPAC learning using Theorem 4.5.

We can use the discreteness of $\Sigma$ to give a bound on the number of distinct functions in $H$. Indeed, there are $|\Sigma|^d$ distinct values of $(x_0, \tau[-l :])$ that would determine the number of distinct behaviors of any $h_{w,w_0} \in H$. By Sauer's lemma, we have $\Gamma_H(s^d) \leq \left(\frac{2es^d}{d+1}\right)^{d+1} = s^{O(d^2)}$. This allows us to use Theorem 4.4 to give a bound of $O\left(\frac{1}{\epsilon}(d^2 \log(s) + \log \frac{1}{\delta})\right)$ on the sample complexity for TVPAC learning that is independent of the length $T$ of the trace.

As an example of a naturally discrete and finite setting, where the problems, the reasoning steps and the verifiers all come from finite sets, consider the following example.

**Example E.3** (Valid reasonings on a graph). In this example, valid reasonings are paths in a graph, part of which is given by $x_0$ and part of which is implicit, defined by an unknown ground-truth verifier $h^*$. Formally, let $G = (V, E)$ denote the complete graph on $n$ nodes. Let $X = V \times 2^E$ and $\Sigma = E$. The verifier class consists of functions

$$H = \{h_{\tilde{E}} : (x_0 = (v_0, E_0), (x_1 = (v_0, v_1), \ldots, x_i = (v_{i-1}, v_i)))$$
$$\mapsto \mathbb{I}[\wedge_{j \in [i]}\{x_j \in E_0 \cup \tilde{E}\}] \mid \tilde{E} \subseteq E\}$$

that verify whether each step $(x_{j-1}, x_j)$ of the reasoning trace is valid, where a valid step is either an edge from $E_0$ specified in the problem $x_0$, or in the (unknown) set of edges $E^*$ corresponding to $h^* = h_{E^*}$. Note that $H$ is intersection-closed and $|H| = 2^{|E|} = 2^{n(n-1)/2}$. The natural approach of building an estimate $\hat{E}$ of $E^*$ by collecting only the edges in the positively labeled traces in the training examples that are not already included in the problem $x_0$ corresponds to the closure algorithm. Therefore, we have SVPAC, TVPAC and $\gamma$-TVPAC learning with $\tilde{O}(n^2/\epsilon)$ sample complexity (using Theorem 3.2, Theorem 4.4, and Corollary D.5).

The above example could be used to model a discrete puzzle like the farmer-fox-chicken-corn puzzle or the sliding tile puzzle. The vertices would correspond to the different discrete states of the puzzle. The final goal state is assumed to be fixed (for example, the sliding tiles make the desired picture), and the problem statement $x_0 = (v_0, \emptyset)$, where $v_0$ is some initial state.

Since one of our main motivations is to learn good verifiers for Chain-of-Thought reasoning, for which Large Language Models (LLMs) have been proposed as good candidate generators, it is natural to try to understand our results for verification of natural language reasoning produced by these generators. In the following example, we suppose that we have a finite collection of $K$ verifiers which are also LLMs.

**Example E.4** (Finite set of LLM verifiers)**.** Let $\mathcal{A}$ denote the (finite) set of tokens in a natural language. Let $X = \Sigma = \mathcal{A}^R$, where $R$ is the maximum number of tokens allowed in a single problem statement or reasoning step. Let $H$ be a collection of $K$ LLM verifiers. Under realizability, our results imply that the sample complexity of learning a verifier with small error is $\tilde{O}\left(\frac{\log K}{\epsilon}\right)$ for SVPAC and TVPAC learning, and $\tilde{O}\left(\frac{K}{(1-\gamma)\epsilon}\right)$ for $\gamma$-TVPAC learning (using Theorem 3.2, Theorem 4.4, and Theorem 4.9 respectively). We show sample complexity bounds without the realizability assumption in Appendix F.

We conclude this section with an example where it is possible to learn a verifier online with a bounded number of mistakes.

**Example E.5.** The problem space is $X = \mathbb{R}^{d \times n}$, that is, each problem $x_0$ consists of a finite number of vectors in $\mathbb{R}^d$. Reasoning steps are also vectors in $\Sigma = \mathbb{R}^d$. $h^*$ is also given by a set of vectors in $\mathbb{R}^d$ (unknown to the learner). For a given problem $x_0$, a reasoning step $x_i$ is said to be valid if it lies in $\mathrm{span}(x_0, h^*)$, the subspace spanned by the problem $x_0$ and the hidden vectors $h^*$, and incorrect otherwise. The verifier is presented by a sequence of problem-reasoning pairs $(x_0^{(1)}, x_1^{(1)}), (x_0^{(2)}, x_1^{(2)}), \ldots$, and gives an assessment YES or NO for each pair. The verifier is said to suffer a mistake if either it accepts a faulty reasoning $x_1^{(i)} \notin \mathrm{span}(x_0^{(i)}, h^*)$, or says NO for a valid reasoning $x_1^{(j)} \in \mathrm{span}(x_0^{(j)}, h^*)$.

First, we make a simplifying assumption that all problem vectors in any problem $x_0$ lie in a space orthogonal to $\mathrm{span}(h^*)$. For this case, we will show an online learner that is sound (i.e. never accepts a faulty reasoning) and makes at most $\dim(\mathrm{span}(h^*)) \leq d$ mistakes. We initialize $h = \{\}$ and will maintain the invariant that $\mathrm{span}(h)$ is a subspace of $\mathrm{span}(h^*)$. Given $(x_0^{(i)}, x_1^{(i)})$, we accept the reasoning if $x_1^{(i)}$ lies in $\mathrm{span}(x_0^{(i)}, h)$, and reject otherwise. Our invariant $\mathrm{span}(h) \subseteq \mathrm{span}(h^*)$ implies that we never accept an invalid reasoning. If we make a mistake on $(x_0^{(i)}, x_1^{(i)})$, then we add the component of $x_1^{(i)}$ orthogonal to $\mathrm{span}(x_0^{(i)}, h)$ (i.e., $x_1^{(i)} - \mathrm{proj}(x_1^{(i)}, \mathrm{span}(x_0^{(i)}, h))$, where $\mathrm{proj}(v, S)$ denotes the projection of vector $v$ onto the subspace $S$) to $h$. This increases $\dim(\mathrm{span}(h))$ by 1 and maintains our invariant $\mathrm{span}(h) \subseteq \mathrm{span}(h^*)$. Therefore, this algorithm makes at most $\dim(\mathrm{span}(h^*)) \leq d$ mistakes.

Next, we show a small mistake bound even when we remove the orthogonality assumption above. Any problem $x_0$ is given by a finite collection of vectors in $\mathbb{R}^d$ as above, and assume that $h^*$ is given by a single vector in $\mathbb{R}^d$. In this case, we will show a mistake bound of $d+1$, but will allow two-sided error (in the previous case, our algorithm never resulted in false positives). Let $S^*$ denote a subspace maintained by the algorithm that has the invariant that it always contains $h^*$. Initialize $S^* = \mathbb{R}^d$. Given a problem $(x_0, x_1)$, we first check if $x_1 \in \mathrm{span}(x_0)$, and return YES if so (which is always correct). Else, we return NO until the first mistake. At this point we set $S^* = \mathrm{span}(x_0, x_1)$. For any new instance $(\overline{x}_0, \overline{x}_1)$, we update $S^*$ upon mistakes. We consider the following cases.

1. $S^* \subseteq \mathrm{span}(\overline{x}_0, \overline{x}_1)$.

   a. $S^* \subseteq \mathrm{span}(\overline{x}_0)$. In this case, $h^* \in \mathrm{span}(\overline{x}_0)$ or $\mathrm{span}(\overline{x}_0, h^*) = \mathrm{span}(\overline{x}_0)$. Thus, it suffices to output YES iff $\overline{x}_1 \in \mathrm{span}(\overline{x}_0)$. We do not make any mistakes in this case.

   b. $S^* \nsubseteq \mathrm{span}(\overline{x}_0)$. In this case, we say YES. Since $h^* \in S^* \subseteq \mathrm{span}(\overline{x}_0, \overline{x}_1)$, we can write $h^* = \overline{a}.\overline{x}_0 + b\overline{x}_1$. If we made a mistake, then $\overline{x}_1 \notin \mathrm{span}(\overline{x}_0, h^*)$. This implies $b = 0$ and $h^* \in \mathrm{span}(\overline{x}_0)$. Thus, we can set $S^*$ to $S^* \cap \mathrm{span}(\overline{x}_0)$. The dimension is reduced by at least one, since we assumed $S^* \nsubseteq \mathrm{span}(\overline{x}_0)$.

2. $S^* \nsubseteq \mathrm{span}(\overline{x}_0, \overline{x}_1)$. In this case, we say $\mathbb{I}[\overline{x}_1 \in \mathrm{span}(\overline{x}_0)]$. We don't make a mistake when we say YES. If we made a mistake, then $\overline{x}_1 \in \mathrm{span}(\overline{x}_0, h^*)$ and $\overline{x}_1 \notin \mathrm{span}(\overline{x}_0)$. This implies $\overline{x}_1 = \overline{a}.\overline{x}_0 + bh^*$ with $b \neq 0$. Therefore, $h^* \in \mathrm{span}(\overline{x}_0, \overline{x}_1)$. Thus, we can safely update $S^*$ to $S^* \cap \mathrm{span}(\overline{x}_0, \overline{x}_1)$, and the dimension of $S^*$ goes down by at least 1.

Thus, $\dim(S^*)$ goes down by 1 every time we make a mistake except possibly for the first time, for a total mistake bound of $d+1$.

# F  Beyond Realizability

The main focus of our work is the realizable case, where a perfect $h^*$ lies in our verifier class $H$ which makes no mistakes on any problem-trace pair (i.e., accepts exactly the right reasoning traces for all problems in $X$). This property is particularly desirable for verification. However, it might be the case that our search space for verifiers is limited and no verifier in $H$ perfectly verifies all the reasoning traces for all the problems of interest. This is known as the *agnostic* setting in PAC learning terminology, and the goal is to learn a verifier $h$ that has error almost as small as the verifier with the smallest error in $H$. Here we will formally define agnostic SVPAC and TVPAC learning and use arguments from standard PAC learning theory to show sample complexity bounds for agnostic learning of verifiers. Note that the corresponding question for Chain-of-Thought generation was left open by prior work [JVB+25].

## F.1  Agnostic simple verifiers

The "label" for a problem-trace pair $(x_0, \tau = (x_1, x_2, ..., x_t))$ is given by $y = (y_1, \ldots, y_t) \in \{YES, NO\}^t$. Given $y \in \{YES, NO\}^T$ let $\mathsf{f}(y)$ denote the smallest index $i \in [T]$ such that $y_i = NO$ (and $\mathsf{f}(y) = T$ if $y_i = YES$ for all $i$). For a verifier $h \in H$ define its loss w.r.t. label $y$ as

$$\ell_h(x, \tau = (x_1, ..., x_T); y = (y_1, \ldots, y_T)) := \mathbb{I}[h(x_0, (x_1, ..., x_j)) \neq y_j] \quad \text{for some } j \leq \mathsf{f}(y).$$

That is, we penalize the verifier for rejecting a trace while it is still correct according to the label $y$, or failing to reject at the first index that the label indicates as faulty (the rest of the label does not matter in this case). Formally, we have the following definition for agnostic learning.

**Definition F.1** (agnostic SVPAC-learnability). Let $X$ denote the problem space and $H \subseteq \{YES, NO\}^{X \times \Sigma^*}$ denote the class of verifiers. Then a learner is said to be an agnostic simply-verifiably-PAC learner for $H$ with sample size $m = M(\epsilon, \delta)$ (sample complexity is the smallest such $m$) if for any $\epsilon, \delta \in (0, 1)$, for any distribution $D$ over $X \times \Sigma^T \times \{YES, NO\}^T$, for $h^* \in \text{argmin}_{h \in H} \mathbb{E}_{(x_0, \tau, y) \sim D}[\ell_h(x, \tau; y)]$, given a sample $S \sim D^m$, the learner outputs a verifier $h$ such that with probability at least $1 - \delta$ over the draw of $S$,

$$\mathbb{E}_{(x_0, \tau, y) \sim D}[\ell_h(x_0, \tau, y) - \ell_{h^*}(x_0, \tau, y)] \leq \epsilon.$$

The learner is said to be proper if $h \in H$.

We now show that it is possible to agnostically SVPAC learn a verifier with small sample complexity for any finite class of verifiers $H$. A simple Hoeffding's bound based argument familiar from standard agnostic PAC learning implies that we can learn a good verifier with $\tilde{O}\left(\frac{1}{\epsilon^2} \log |H|\right)$ labeled problem-trace samples.

**Theorem F.2.** Any finite class of verifiers $H$ is agnostically SVPAC-learnable with sample complexity $O\left(\frac{1}{\epsilon^2}(\log(|H|) + \log \frac{1}{\delta})\right)$.

*Proof.* We use ERM, i.e. simply output any verifier $\hat{h} \in H$ that achieves the smallest total loss $\ell_h$ on the training sample and show that it achieves the stated sample complexity. Since the examples in the training sample $S$ are iid draws from $D$, the loss of a fixed $h$ on the examples is an iid $\{0, 1\}$-valued variable. By Hoeffding's bound,

$$\Pr\left[\left|\mathbb{E}_D[\ell_h(x, \tau; y)] - \frac{1}{|S|} \sum_{(x^{(i)}, \tau^{(i)}, y^{(i)}) \in S} \ell_h(x^{(i)}, \tau^{(i)}; y^{(i)})\right| \geq \frac{\epsilon}{2}\right] \leq 2e^{\frac{-|S|\epsilon^2}{2}}.$$

By a union bound,

$$\Pr\left[\exists h \in H \text{ s.t. } \left|\mathbb{E}_D[\ell_h(x, \tau; y)] - \frac{1}{|S|} \sum_{(x^{(i)}, \tau^{(i)}, y^{(i)}) \in S} \ell_h(x^{(i)}, \tau^{(i)}; y^{(i)})\right| \geq \frac{\epsilon}{2}\right] \leq 2|H|e^{\frac{-|S|\epsilon^2}{2}}.$$

Applying this to ERM $\hat{h}$ and $h^*$, and noting that the error of $\hat{h}$ on $S$ is no larger than that of $h^*$, implies that

$$\mathbb{E}_{(x_0, \tau, y) \sim D}[\ell_{\hat{h}}(x_0, \tau, y) - \ell_{h^*}(x_0, \tau, y)] \leq \epsilon,$$

with failure probability $\delta \leq 2|H|e^{\frac{-|S|\epsilon^2}{2}}$. Solving for $|S|$ gives the desired bound. $\square$

Since our proof for Theorem 3.3 involves bounding the relevant shattering coefficient, we can also readily adapt the proof of the fundamental theorem of PAC learning to establish a $\tilde{O}(\frac{1}{\epsilon^2}\mathsf{VCDim}(H)\log T)$ bound on the sample complexity of agnostic SVPAC-learning for verifier classes $H$ with a finite VC dimension.

## F.2 Agnostic trustable verifiers

We give a similar agnostic extension for TVPAC learning where the learner has access to a gold standard reasoner that provides up to $k$ correct reasoning traces for any problem $x \in X$, and when $\Sigma$ is finite. For a verifier $h$, we denote its population error as

$$\mathrm{err}_D(h) := 1 - \Pr_{x \sim D}[h \text{ is 1-complete w.r.t. } g \text{ and sound for } x].$$

**Definition F.3** (agnostic TVPAC-learnability). Let $X$ denote the problem space and $H \subseteq \{\mathsf{YES}, \mathsf{NO}\}^{X \times \Sigma^*}$ denote the class of verifiers. Let $g(x) \subseteq \Sigma^T$ denote the set of correct reasoning traces for any $x \in X$. Then a learner is said to be an agnostic trustably-verifiably-PAC learner for $H$ with sample size $m = M(\epsilon, \delta)$ (sample complexity is the smallest such $m$) if for any $\epsilon, \delta \in (0, 1)$, for any distribution $D$ over $X$, for $h^* \in \operatorname{argmin}_{h \in H} \mathrm{err}_D(h)$ and $\mathsf{OPT} = \mathrm{err}_D(h^*)$, given a sample $S \sim D^m$ and for each $x \in S$ given access to the set $g(x)$, the learner outputs a verifier $h$ such that with probability at least $1 - \delta$ over the draw of $S$, $\mathrm{err}_D(h) \leq \mathsf{OPT} + \epsilon$. The learner is said to be proper if $h \in H$.

We show that ERM on the samples constructed using the gold standard reasoner in Section 4.1 is an agnostic SVPAC learner with small sample complexity for any finite class of verifiers $H$. The argument is similar to that of Theorem F.2.

**Theorem F.4.** Any finite class of verifiers $H$ is agnostically TVPAC-learnable with sample complexity $O\left(\frac{1}{\epsilon^2}(\log(|H|) + \log\frac{1}{\delta})\right)$.

*Proof.* The key observation is that our training sample $S = (x^{(i)}, g(x^{(i)}))_{i \in [m]}$ allows us to determine $\mathbb{I}[h \text{ is 1-complete w.r.t. } g \text{ and sound for } x]$ for any problem $x$ in the sample, by using the tree $\mathcal{T}_g(x)$ and finiteness of $\Sigma$. This gives us the 0-1 loss of $h$ on $x$ which can be used to implement the ERM, and we can apply the same argument as in the proof of Theorem F.2 for this loss to conclude the proof. $\qquad\square$

As before, we can use the bound on the shattering coefficient in our proof of Theorem 4.5 and adapt the proof of the fundamental theorem of PAC learning to establish a $\tilde{O}(\frac{1}{\epsilon^2}\mathsf{VCDim}(H)\log kT|\Sigma|)$ bound on the sample complexity of agnostic TVPAC-learning for verifier classes $H$ with a finite VC dimension.

