# OpenReview forum: "On Learning Verifiers and Implications to Chain-of-Thought Reasoning"
_NeurIPS.cc/2025/Conference — NeurIPS 2025 poster_

### Official Review · Reviewer_1Yc9 · 2025-06-24

**Clarity:** 2
**Significance:** 2
**Originality:** 3
**Rating:** 4
**Confidence:** 3

**Summary:**

The authors present a series of analyses and theoretical results on the PAC-learnability of correct verifiers for step-by-step sequential reasoning, a process that has become increasingly popular in the application of LLMs. The authors' key results show that you can PAC-learn a consistent in-disttribution (with respect to some known prior) verifier with a log-bounded sample complexity, and present trustable verifiers where accepted proofs must come from a known and fully correct line of reasoning. These trustable verifiers, while more challenging to PAC-learn, do have a lower bound on sample complexity if the number of gold standard proofs is small.

**Questions:**

Questions:
* is there not a relationship between this work and LLMs that use interactive theorem provers that also ensure the soundness and correctness of intermediate steps? can the authors comment on what their work can do that others, such as LeanDojo?
* Line 158: Can you comment on what distributional assumptions we can reasonably make when actually dealing with certain classes of LLM-guided reasoning problems? is there other work on such analysis?
* How reasonable is the gold-standard assumption in lines 163-164? how does this affect the tractability of the overall approach? More concretely, in what kinds of problem settings do we expect this assumption to be valid and when do we not?
* line 186: does a satisfactory learner exist for all values of $\eta$ greater than zero? does $\eta$ exist for sample efficiency reasons or is it a necessary assumption?
* Line 190: why do we make this assumption?

**Ethical Concerns:**

["NO or VERY MINOR ethics concerns only"]

**Final Justification:**

My score was predicated on the changes that were promised by the authors in the rebuttal; as such, I will maintain my score.

**Limitations:**

The assumptions necessary for the theory to hold are generally laid out but lack a bit of detail regarding their relative strength in the main text.

**Quality:**

3

**Strengths And Weaknesses:**

Strengths:

* Achieving PAC-bounds and formal guarantees on the correctness and achievability of reasoning is a very very valuable problem to look at and a nice step forward from an algorithmic perspective.
* The formulation of the problem setting is done in a rather understandable way and the key results are accompanied by intuition and explanation to ground the reader. It’s a nice and approachable way to write a theory paper for the ML community.
* The results and commentary on the results do provide some interesting insights towards LLM-guided sequential reasoning. In particular, I found Theorem 4.4 and Theorem 4.8 particularly compelling in terms of their implications for how easy it will be to learn verifiers for LLMs at scale.
* The writing in section 4 (the bulk of the paper) is fairly easy to follow. The theoretical results are compelling despite the assumptions made.

Weaknesses:

* The writing in the introduction, in my opinion, doesn’t do a good enough job of making the problem statement, motivation, and overall contribution clear. Paragraph 1 is light on detail and doesn’t make the scope immediately clear or tight. Later on in the introduction, it is also unclear what the exact contribution of the work is (paragraph 4, line 39.) if there are many correct reasoning traces starting from x0 why not accept all of them? what exactly is the difference between a ‘gold standard’ correct reasoning trace and some other correct reasoning trace? These questions are answered in section 4 but there needs to be a more rigorous explanation of the key results in the introduction to properly ground the reader.
* The related work section feels rather light - there has been a huge amount of effort in auto formalization, formal reasoning with LLMs, verification of LLMs both in a formal methods-y and not formal methods-y sense, and so forth. The authors would be well suited to make a more substantial effort in contextualizing their work in the large amount of recent work to better situate their contribution.
* It’s broadly unclear to me how this work is specific in any capacity to LLMs, besides the fact that LLMs are being used to generate step-by-step reasoning based proofs and the work applies to this problem setting. The actual usage of LLMs doesn’t actually appear in the paper whatsoever and it begs questions of relevance towards its claimed application. That is not to say that the theoretical contributions aren’t of interest to the community in general - they certainly are, but the paper is written with the central motivation coming from LLM applications. The paper doesn’t however actually apply the theory to LLMs at all. So either the framing of the story ought to change or there need to be results that actually demonstrate the efficacy and reasonableness of the theory by applying it to LLM-guided formal reasoning.
* A number of assumptions are made throughout the paper to facilitate the theory. This is fine in a nutshell, but there is little to no commentary on (1) why these assumptions are made in the context of the theory, or (2) whether or not these assumptions are reasonable when actually performing LLM-guided formal reasoning. I’ve asked about some of them in my questions.

    * Line 240 provides an interesting insight but is not immediately followable from the preceding theory.

Overall I think the paper presents a nice contribution but the writing and framing of the story causes consternation on my end. I will recommend acceptance but would suggest the authors reframe the story a bit to match their contribution.

---

> ### Author Rebuttal · Authors · 2025-07-31
>
> We thank the reviewer for their detailed review. We also appreciate their comments that our paper is approachable, and that our results provide compelling insights for a valuable problem. We provide our response to the reviewer’s comments and questions below.
>
>
> We propose the following updates to improve our writing based on the suggestions in the review.
>
> **Introduction**. We propose to add the following list of contributions to make our problem statement and overall contribution clear.
>
> • We introduce a formal framework for studying verifiers for Chain of Thought reasoning. Given any problem statement and a sequence of reasoning steps for the problem, we propose the problem of learning verifiers that examine the steps for correctness, and for an incorrect reasoning trace return the first faulty step in the reasoning.
>
> • We formally define simple verifiers which have access to random Chain of Thought reasoning sequences labeled as “correct” or “incorrect” along with the first faulty step. We establish sample complexity bounds for learning good simple verifiers in a PAC sense for verifier classes that are finite or have a finite VC dimension.
>
> • We next introduce the more powerful trustable verifiers, that only have access to random problems and a gold standard reasoner that provides a small number of guaranteed correct reasoning traces for each sampled problem. We establish PAC learnability of designing verifiers that accept all the gold standard reasoning traces on most problems and never accept faulty reasoning traces, provided the space of reasoning steps is finite.
>
> • Finally, we extend our trustable verification goal to the case where there may be a large number of gold standard reasoning traces, but only a random correct trace is available to the learner. We establish upper and lower bounds on the sample complexity of learning a verifier that is always sound (i.e., never accepts an incorrect trace) and accepts most of the gold standard traces on most problems.
>
>
> **Related Work**. We will significantly expand on our related work section to discuss more key works related to chain-of-thought generation and verification. This would include theoretical works that study verifier-assisted generation e.g. Amit et al. (2024), Botta et al. (2025), as well as more applied works e.g. Nakano et al. (2021), Hao et al. (2023), etc.
>
> **Overall framing**.  We will be happy to change the title to something like “On learning verifiers for stepwise reasoning” to reflect the generality of our work beyond chain-of-thought reasoning by Large Language Models (even though it is currently the most relevant reason to study this).
>
> **Discussion of assumptions and implications**. Our assumptions mainly correspond to different verification goals that one might have.
>
> 1. Simple verification (SVPAC). Here the training data consists of problem statement and reasoning-sequence pairs. This could be generated for example by asking an LLM to solve some randomly drawn problems while “showing its reasoning”. This is similar to the training data for chain-of-thought generation studied in prior theoretical works [Joshi et al. (COLT 2025), Malach (ICML 2024)]. For learning a verifier, we assume these are annotated as either correct or with the first incorrect reasoning step.
>
> 2. Trustable verification (TVPAC). Here our training set consists of problem statements along with a small number of “gold standard reasoning” steps that solve each problem. For example, this could be a set of homework problems in a graduate course and the solutions provided by the instructor. The goal of the verifier is to verify whether the reasoning matches one of the standard solutions (even in unseen problems and reasoning sequences), and work correctly (only accept correct proofs) for most typical problems.
>
> 3. A stronger trustable verification model ($\gamma$-TVPAC). Here our training set consists of problem statements along with one correct sequence of reasoning steps that solves each problem. The goal of the verifier is to verify whether the reasoning is correct (even in unseen problems and reasoning sequences), for most typical problems. Our verifier is sound (never accepts a wrong reasoning) and almost complete (accepts nearly all typical correct reasonings).
>
>
>
>
> **Relation with other works that use "interactive theorem provers"**: Theoretical work on using interactive theorem provers e.g. Amit et al. (2024) assume a sound and complete verifier as given, and show how the reasoning generator can use this verifier to improve their correctness. In contrast, we study how to actually learn these verifiers from data.
>
> LeanDojo extracts proofs in the functional programming language Lean into datasets for training machine learning models. It also enables the trained model to prove theorems by interacting with Lean's proof environment. Note that the “verifier” in this interaction is the Lean proof system, which only accepts formal language and prior applied work (e.g. Yang et al., NeurIPS 2023) has tried to use language models to write proofs in the Lean system (while they also generate natural language descriptions of their proof steps, the evaluation is through correctness of the lean proof). In contrast, our work studies the learnability of verifiers that potentially evaluate natural language reasoning directly.
>
>
> **On distributional assumptions [Line 158]**
>
> In our trustable verification model, we only assume that the problem statements are coming from some distribution but the reasoning traces that we verify may be arbitrary (and out-of-distribution with respect to the training set). This roughly corresponds to building verifiers that are trustable for problems coming from a specific application domain (e.g. solving math problems or logic puzzles), in the sense that they are sound and (nearly) complete.
>
>
> **On gold standard assumption [Line 163-164]**
> *“Line 190: why do we make this assumption?”* [the number of gold standard reasonings is small]
>
> Discrete puzzles like the farmer-fox-chicken-corn puzzle may have one or a small number of correct solutions. Another example is a set of homework problems in a graduate course and the gold standard solutions provided by the instructor. Verifying all (potentially out-of-distribution) proofs for unseen problems and proofs from a given domain is a more challenging goal than simple verification (Section 3), but under the gold standard verification goal we only aim for completeness with respect to the gold standard proofs (but still always sound).
>
> [Line 190] The main reason for assuming the number $k$ of gold standard reasonings to be small is for computational efficiency (linear in $k$) and sample efficiency (logarithmic in $k$) of learning verifiers.
>
> We further relax this assumption in our strongest verification goal of $\gamma$-TVPAC learning (Section 4.2), but have stronger impossibility results for this goal (Theorems 4.8, 4.9). So the gold standard assumption is a nice intermediate setting, where we can get robust verification with small sample complexity.
>
>
> *“line 186: does a satisfactory learner exist for all values of η greater than zero? does η  exist for sample efficiency reasons or is it a necessary assumption?”*
>
> Yes, there is a learner for each value of $\eta$ greater than zero (the sample complexity upper bound depends on $\eta$) and our sample efficiency depends inversely on $\eta$ (Theorem 4.7). Perfect completeness ($\eta = 0$) is achieved under the gold standard assumption (Theorems 4.3, 4.4).
>
>
>
>
>
>
>
>
>
>
>
>
>
>
>
> **References**
>
>
> [1] Amit, Noga, Shafi Goldwasser, Orr Paradise, and Guy N. Rothblum. "Models That Prove Their Own Correctness." In ICML 2024 Workshop on Theoretical Foundations of Foundation Models.
> [2] Botta, Edoardo, Yuchen Li, Aashay Mehta, Jordan T. Ash, Cyril Zhang, and Andrej Risteski. "On the Query Complexity of Verifier-Assisted Language Generation." In Forty-second International Conference on Machine Learning 2025.
> [3] Hao, Shibo, Yi Gu, Haodi Ma, Joshua Jiahua Hong, Zhen Wang, Daisy Zhe Wang, and Zhiting Hu. "Reasoning with Language Model is Planning with World Model." In The 2023 Conference on Empirical Methods in Natural Language Processing.
> [4] Nakano, Reiichiro, Jacob Hilton, Suchir Balaji, Jeff Wu, Long Ouyang, Christina Kim, Christopher Hesse et al. "Webgpt: Browser-assisted question-answering with human feedback." arXiv preprint arXiv:2112.09332 (2021).
> [5] Yang, Kaiyu, Aidan Swope, Alex Gu, Rahul Chalamala, Peiyang Song, Shixing Yu, Saad Godil, Ryan J. Prenger, and Animashree Anandkumar. "Leandojo: Theorem proving with retrieval-augmented language models." Advances in Neural Information Processing Systems 36 (2023): 21573-21612.

---

> > ### Comment · Reviewer_1Yc9 · 2025-08-04
> > **Thanks for your response**
> >
> > Thank you to the authors for the detailed response.
> >
> > The additional discussions mentioned in the rebuttal help clarify the overall contribution and contextualization significantly. I encourage the authors to include all of the discussed changes in the final version of the paper (although I don't think the title needs to change too much.)

---

### Official Review · Reviewer_oW28 · 2025-07-02

**Clarity:** 4
**Significance:** 3
**Originality:** 4
**Rating:** 5
**Confidence:** 5

**Summary:**

This is a theoretical paper that analyzes verifiers' computational complexity. It mostly concerns the standard bounds using the Vapnik-Vitanyi framework for PAC theory; the novelty is in considering the soundness and completeness of a verifier. The natural application is to validate CoT reasoning in LRM. One important conclusion is given for strong completeness (learning verifiers accepting all correct reasoning and refusing all incorrect ones). Without Oracle, the sample grows prohibitively with the reasoning alphabet size and (maximum) trace length. Key insight is that, under normal conditions, learning robust and trustworthy (trustable) CoT verifiers is fundamentally constrained.

**Questions:**

1. Given the impossibility of an effective CoT verifier with sufficiently many traces, what implication would one expect for trustable/trustworthy verifiers? Perhaps, you can elaborate on this in the paper.
2. Given the impossibility of a sound and complete verifier in an infinite space, what feasible recommendations can be provided for designing a robust LRM? The recommendation can point to problems and closed systems where  CoT is proven efficient and/or practical (e.g., problem solving for Euclidean geometry, medium-complexity math problems, code generation (with limited verification), etc.
3. An interesting research question is how to describe domains admitting a trusted verifier that almost surely gives a correct answer to a new question.

**Ethical Concerns:**

["NO or VERY MINOR ethics concerns only"]

**Final Justification:**

Recommended score: 5 (accept). The issues raised by reviewers have been resolved, in my opinion.

The paper opens up a promising direction in understanding the limitations of LRM reasoning, with the author's rebuttal on the mathematical context and the Math benchmark GSM8K.

The natural next step for further theoretical and experimental research is benchmarking for novel problems (cf. Benchmarking LLMs on Advanced Mathematical Reasoning, the Berkeley report No. UCB/EECS-2025-121 by J. Yue and D. Klein), and FrontierMath (Glazer et al., 2024). Of particular interest are benchmarks based upon the formal languages (Lean) because they provide a verifier with an opportunity to self-check the validity of each step in CoT.

That leads to a trustworthy (trustable) verification. The paper establishes a bound for sample complexity when the number of correct samples $\textit{k}$ is small: loosely, a function of $\Omega(log|\Sigma|)$, while for large $\textit{k}$, verification requires $\Omega(|H|)$ samples ($H$ is a hypothesis class, and $\Sigma$ is the set of reasoning steps). These are important practical limitations for the scalability of trustable verification.

**Quality:**

3

**Strengths And Weaknesses:**

Originality.
A useful view of the problem is simple verifiers vs. trustable verification within a formal framework. Moreover, this paper establishes the complexity boundaries for these cases.
The approach associating the completeness and soundness of a classifier/verifier with its complexity is original. However, the terminology may be a little bit confusing in this context since it clashes with the well-known and established notions from standard logic.  One can feasibly talk about classifiers and verifiers within their domain's first-order logic. The completeness and soundness would mean something else.

Significance.
The bounds and key limitations are significant and correlate with recent results in automated proving. The results provide an additional view and framework for tracing the complexity of CoT verifiers. This, in turn, generates some interesting research questions for this key area of LRM (the authors posed some implicitly).
The reasoning alphabet for the CoT scenario is infinite since there is proof of any length in the space. We cannot generalize the results of this paper to this case. And for trustable verification, validating almost surely requires samples of unbounded complexity (due to no free lunch, and other classic considerations).
However, it is easy to see examples of the TVPAC classifiers with infinite VC-dimension (if we relax the PAC requirement for TVPAC). Check one that solves the problem, whether a randomly selected number is transcendental.   Almost surely, classifying it, in all cases, would constitute a PVAC classifier regardless of sampling size.

Is it possible to get close to that kind of behavior for the PAC case, i.e., for the classifier with finite VC dimension depending on overall complexity?
Limitations of the framework: The authors claim, "All limitations have been discussed in context". They are among the key results. Thus, the limitations are properly addressed.

---

> ### Author Rebuttal · Authors · 2025-07-31
>
> We thank the reviewer for their time and their appreciation of our contributions.
>
>
>
> *“One can feasibly talk about classifiers and verifiers within their domain's first-order logic.”*
>
> This is a great point. While we clearly distinguish our work from formal verification in the introduction and related work (we learn verifiers for natural language instead of formal language), we will make sure to clarify the differences in our terminology by adding the following remark near our definitions of soundness/completeness.
>
> We note that soundness and completeness of proof systems is a terminology also used in formal verification and logic, and caution the reader from conflating them with our notions.  A soundness guarantee for a deductive system expresses that all provable sentences are true. Completeness states that all true sentences are provable. While there is an analogy, we remind the reader that our study applies to natural language reasoning while formal logic involves proofs expressed in a very precise formal language and their verification.
>
>
> *“Given the impossibility of an effective CoT verifier with sufficiently many traces, what implication would one expect for trustable/trustworthy verifiers”*
>
>
> Our impossibility results (in our strongest verification model of $\gamma$-TVPAC learning) set up a situation where there are many different “types” of proof (think of each $S_i$ in our proof as a different proof type), some of which are legitimate and some of which are not, and where seeing some legitimate types gives no information about which are not.  So our results suggest that situations of this kind may require a large amount of training data.
>
>
> Our theory suggests another way around in the absence of large amounts of data. One may relax the verification goals to a trustable verifier that is sound, but only complete with respect to a small set of gold standard reasoners (Section 4.1) where the sample complexity upper bounds are more reasonable.
>
>
>
>
> *“Given the impossibility of a sound and complete verifier in an infinite space, what feasible recommendations can be provided for designing a robust LRM?”*
>
> This is an interesting question. We remark (lines 318-325) that one way to overcome our impossibility results (Theorems 4.8, 4.9) for building a trustable verifier is to restrict the domain to intersection-closed verifier classes (which may be infinite). An interesting example is noted in Appendix C (Example C.3) where this intersection-closed property holds — here we use a graph-based construction to model discrete puzzles (e.g. the farmer-fox-chicken-corn puzzle or the sliding tiles puzzle) where nodes of the graph can denote different states of the puzzle.
>
> But this is only one example of a sufficient condition for infinite classes and one example of a domain where sound and complete verification is possible. An interesting direction for future research (as you note) is to characterize exactly when trustable verification is possible (necessary and sufficient conditions).
>
>
>
>
> *“An interesting research question is how to describe domains admitting a trusted verifier that almost surely gives a correct answer to a new question.”*
>
>
> We agree with the reviewer that this is a very interesting question for future research.

---

> > ### Comment · Reviewer_oW28 · 2025-08-05
> > **Thank you for your prompt response**
> >
> > I would like to emphasize the practical usefulness and importance of your approach in the rigorous context. Although this is a first step in this direction, trustworthy (trustable) verification adds formal guarantees. Certainly, this also leads to natural limitations, e.g., stricter data and computational demands, which is inevitable trade-off.
> > Within the framework, the impossibility (infeasibility of $\gamma$-verifier) for trustable verification with large $k$ (due to the sample complexity for learning, $\Omega(|H |)$), is one of the important limitations. It indicates fundamental boundaries of CoT, including in formal settings. In my opinion, it is a promising direction of research.

---

### Official Review · Reviewer_9kqw · 2025-07-02

**Clarity:** 1
**Significance:** 2
**Originality:** 3
**Rating:** 4
**Confidence:** 2

**Summary:**

The paper tackles the problem of learning a verifier that can check whether a given sentence in natural language is correct in the sense that the reasoning steps it takes are correct according to the distribution used to learn the verifier. The problem is formulated in a PAC setting making the verification thereby probabilistic.

**Questions:**

1) It is not very clear what you mean by chain-of-thought. Because when you introduce chain-of-though you cite the papers that introduced chain of thought prompting. However, none of prompting plays any role in any of the proofs. Could you clarify this?

2) Could you comment on my (possible wrong observation) that your proofs have nothing to do with chain of though (prompting)?

3) Could you comment on the availability of actual data to train such a verifier. For the different scenarios, i.e. for the different flavors of verifiers.

**Ethical Concerns:**

["NO or VERY MINOR ethics concerns only"]

**Final Justification:**

The authors have proposed to change the framing of the paper by moving away from "chain of thought reasoning" and focusing more on learning verifiers of sequences. Of course with the use LLMs as sequence models for language as a main application.

I also appreciate the authors addressing my concern regarding availability of data and grounding their claim in the literature.

I am increasing my score to a "borderline accept".

**Limitations:**

yes

**Paper Formatting Concerns:**

seems good.

**Quality:**

3

**Strengths And Weaknesses:**

The title of the paper is "on learning verifiers for chain-of-thought reasoning". However, it seems that nowhere in the proofs chain-of-thought is actually used as an assumption. If I understand things correctly, is that we have a data set with a sequence of symbols, we then learn a verifier on this set of sequences. After having learned the verifier we check whether a new sequence is from the same distribution as that of the dataset. This happens in Section 3. The remainder of the paper are then variations of this idea.

To me it seems that this has, on a theoretical basis, nothing to do with chain-of-thought reasoning. One could argue that verifying that a sequence of symbols produces by a transformer-based LLM (or any other autoregressive model for that matter) is an application of this sequence PAC learning approach.

An other problem I have is that the paper makes many assumption on availability of data and how the data is given. For instance, the data is assumed to be given on an almost sub-sentence level. This seems like a rather strong assumption. Also, in case that chain-of-thought plays indeed a role in the proposed method (contrary to what I understand) then this would also mess up certain things, as the LLM is seeing only tokens and not these sub-sentence entities.

---

> ### Author Rebuttal · Authors · 2025-07-31
>
> We thank the reviewer for their time and address their concerns below.
>
>
> **Clarification of “chain-of-thought” terminology**: You are right that our usage of chain-of-thought is different from that in the original paper, where it is used primarily as a prompting technique. We use CoT to refer to the standard generation pattern of reasoning models like o3, Deepseek R1, i.e., the model generates the ordered list of intermediate reasoning steps $(s_{1},\dots ,s_{T})$ before its final answer.
>
> Our use of the terminology chain-of-thought is consistent with prior theoretical work e.g. Joshi et al. (COLT 2025), Malach (ICML 2024) which studies chain-of-thought generation (in contrast, we study verification of the reasoning produced by such models). Suppose the language model is given some input question, and it produces an output consisting of intermediate steps (we discuss these steps below in response to your other question) before arriving at the final answer. This behavior may be due to the nature of training data with such chain-of-thought sequences as studied in the prior works mentioned above, or because the language model was prompted to “think step by step”. This nature could also be elicited or amplified by the use of reinforcement learning. As with these theoretical papers on chain-of-thought generation, the study of how to do the chain-of-thought “prompting” itself (which is more related to the ability of the language-model to do in-context learning) is beyond the scope of our work, the applied works we cite indicate challenges with this approach and motivate the need for verification.
>
>
> We will add the above discussion to our paper to clarify the exact sense in which we study chain-of-thought. We will also update our title to “On learning verifiers for *stepwise* reasoning” to reflect the generality of our work beyond the reasoning pattern generated by Large Language Models (even though it is currently the most relevant reason to study this).
>
>
>
>
> **Clarification related to data format**: We describe below the format of training data for the different verification goals studied in our paper.
>
> 1. Simple verification (SVPAC). Here the training data consists of problem statement and reasoning-sequence pairs. This could be generated for example by asking an LLM to solve some randomly drawn problems while “showing its reasoning”. This is similar to the training data for chain-of-thought generation studied in prior theoretical works [Joshi et al. (COLT 2025), Malach (ICML 2024)]. For learning a verifier, we assume these are annotated as either correct or with the first incorrect reasoning step.
>
> 2. Trustable verification (TVPAC). Here our training set consists of problem statements along with a small number of “gold standard reasoning” steps that solve each problem. For example, this could be a set of homework problems in a graduate course and the solutions provided by the instructor. The goal of the verifier is to verify whether the reasoning matches one of the standard solutions (even in unseen problems and reasoning sequences), and work correctly (only accept correct proofs) for most typical problems.
>
> 3. A stronger trustable verification model ($\gamma$-TVPAC). Here our training set consists of problem statements along with one correct sequence of reasoning steps that solves each problem. The goal of the verifier is to verify whether the reasoning is correct (even in unseen problems and reasoning sequences), for most typical problems. Our verifier is sound (never accepts a wrong reasoning) and almost complete (accepts nearly all typical correct reasonings).
>
>
> *“ the data is assumed to be given on an almost sub-sentence level… LLM is seeing only tokens and not these sub-sentence entities.”*
>
> Our theory does not make any particular assumption on the data (only that the reasoning steps come from some set $\Sigma$ which could be anything). The use of an abstract set $\Sigma$ to denote the reasoning steps is consistent with prior theoretical work on chain-of-thought. A natural setting for verification however would be to think of $\Sigma$ as consisting of a reasoning step in natural language. Note that we are learning verifiers here, so even if the LLM that generated the chain-of-thought does next-token prediction, we may ask it to separate its reasoning into “steps” that can be individually verified by the verifier.

---

> > ### Author Response · Authors · 2025-08-05
> > **Author-reviewer discussion**
> >
> > Dear reviewer,
> >
> > We hope our response above clarifies the connection of our setup to the Chain-of-Thought literature and the data format for the different verification goals that we study. We are happy to answer any follow-up questions from the reviewer.
> >
> > Best,
> > Authors

---

### Official Review · Reviewer_ANSN · 2025-07-03

**Clarity:** 2
**Significance:** 2
**Originality:** 3
**Rating:** 3
**Confidence:** 2

**Summary:**

This paper studies the sample complexity of learning verifiers for CoT reasoning in LLMs. The presented formulation of PAC learning on two types of problems, learning a weak verifier and a strong verifier, shows the paper's assumptions and derivation of the bounds.

**Questions:**

Could you revisit two cases in this paper, the weak and the strong verifiers, and explain how the assumption fits the CoT reasoning in LLMs?
For example, we could imagine verifying a reasoning step in a common math reasoning problem.
It seems decomposing a CoT thought into steps is not an obvious task, but assuming that the step is given well, what would the dataset D and the problem set S look like? According to the paper, the reasoning steps are sampled from the entire dataset, and some steps are correct while others are incorrect. Does it imply that the steps can be applied to multiple problems and they can still be correct?
The method also assumes that the labels for the correctness of each step are available. The strong verifier has access to the gold standard reasoner. What would be a rough estimate for |H| in terms of the size of the dataset and the number of steps?

**Ethical Concerns:**

["NO or VERY MINOR ethics concerns only"]

**Final Justification:**

From the rebuttal, I think this paper is a fancy theory paper that provides the bounds and sample complexity results in statistical learning theory. I am not an expert in theory, so this would be a good paper.

One thing that I wish to clarify was the connection to the actual situations by using concrete examples like GSM8K. However, I couldn't find any strong connection to the real use case. I felt some detachment between the theory and practice.

I will keep the current score.

**Limitations:**

YES

**Quality:**

3

**Strengths And Weaknesses:**

**Strength**
The formalization of the learning verifier for LLM reasoning.
This paper will provide a good theoretical basis for learning verifiers.

**Weakness**
There's no discussion on the validity of the assumptions made.
The result is bound, and it avoids being grounded in concrete situations.
It is not apparent whether the assumptions made in this paper apply to CoT reasoning in LLMs.

---

> ### Author Rebuttal · Authors · 2025-07-31
>
> We thank the reviewer for their time spent reading our paper and we address their various questions and concerns below. In particular, we provide simplified discussions of our models (which we are happy to add to the paper) and also point out that we do have concrete examples illustrating our theoretical results in the supplement which the reviewer might have missed.
>
>
> **Concrete examples**: We have included some simple examples in the main body (Section 2, lines 322-325) and also provide several detailed concrete examples for instantiating our theoretical results in Appendix C (Examples C.1 to C.5) in the “Supplementary Material” due to space constraints. We are happy to take the reviewer’s feedback into account and move more examples into the main body in the final version.
>
>
> **Connection to CoT (Chain-of-Thought)**: The decomposition of the reasoning into steps has been previously studied in the literature on CoT generation e.g. Joshi et al. (COLT 2025), Malach (ICML 2024), so it is natural to study it for CoT verification as well. We illustrate below our weak and strong verification goals in the context of CoT reasoning for solving math problems.
>
>
> *Weak (simple) verifiers*. Each problem statement here will be some math problem. The distribution (and unlabeled sample from it) would correspond to problems coming from some domain (e.g. high school, graduate course, international mathematics olympiad, etc.) along with step-wise solutions generated by a CoT LLM reasoner. A labeled dataset will be annotated with verifications of these solutions, either marked correct or the first incorrect step is indicated (this could be done by a human labeler). A learned verifier (potentially another LLM) will use this dataset to learn how to verify unseen (problem+solution) pairs from the same distribution (e.g. more problems from the same domain solved by the same CoT reasoner).
>
> *Strong (trustable) verifiers*. Each problem statement is again (say) some math problem. The distribution (or unlabeled sample) here would correspond to problems coming from some domain (e.g. grad course, olympiad, etc.), but without any solutions. In a labeled dataset, each problem will be annotated with either a small number of correct “gold standard” solutions (say solution key by the problem setter, Section 4.1), or one random correct solution (possibly auto-generated but known to be correct, Section 4.2). A learned verifier (potentially another LLM) will use this dataset to learn how to verify unseen (problem+solution) pairs, with the problems coming from the same distribution but the solutions could be generated by *any* CoT reasoner (LLM in the wild, student-written proof, etc.)
>
> We will be happy to update our title to “On learning verifiers for *stepwise* reasoning” to reflect the generality of our work beyond the reasoning pattern generated by Large Language Models (even though it is currently the most relevant reason to study this).
>
>
> *“According to the paper, the reasoning steps are sampled from the entire dataset, and some steps are correct while others are incorrect. Does it imply that the steps can be applied to multiple problems and they can still be correct?”*
>
>
> We would like to clarify that as explained above, we either sample (problems+reasoning steps pairs) in the simple model; or only the problems in the stronger model (where reasoning steps can be arbitrary). We focus on the realizable setting where the correctness is given by an unknown verifier $h^{\ast}$ and depends not only on the current step but also the problem statement and the sequence of steps so far (recall verifiers in $H$ are functions from $X\times\Sigma^\ast\rightarrow \{\text{YES, NO}\}$). We go beyond the realizable setting in Appendix D where we study the sample complexity of agnostic learning of verifiers.
>
>
> *“What would be a rough estimate for |H| in terms of the size of the dataset and the number of steps?”*
>
>
> Note that we establish upper and lower bounds on the sample complexity i.e. the size of the dataset need to learn a verifier class H in terms of the size |H| of the verifier class (or VC dimension if H is infinite) and the number of reasoning steps T (in cases it impacts sample complexity). This size is logarithmic or linear in |H| depending on the strength of the verification goal and properties of H. For example, if H is intersection-closed, then all our verification goals can be achieved using dataset that is logarithmic in |H|.
>
>
> In terms of a rough estimate for |H|, this would depend on the application; for example, if we are considering verifiers that can be created by fine-tuning a given initial suboptimal verifier, then |H| would be the number of ways this initial verifier could be fine-tuned (say, number of possible weights in the final layer).

---

> > ### Author Response · Authors · 2025-08-05
> > **Author-reviewer discussion**
> >
> > Dear reviewer,
> >
> > We hope our response above clarifies the connection of our setup to the Chain-of-Thought literature and the basic idea underlying the different verification goals that we study. We are happy to answer any follow-up questions from the reviewer.
> >
> > Best,
> > Authors

---

> > > ### Author Response · Authors · 2025-08-07
> > > **Brief reiteration in terms of connection to concrete situations**
> > >
> > > As the author-reviewer discussion period draws to a close, we would like to reiterate to the reviewer that the theory developed in this work captures the essential points and improves the fundamental understanding relevant to the highly active practical area of using verifiers for LLM-generated proofs (e.g. *Prover-Verifier Games improve legibility of LLM outputs*, OpenAI [2024], *Gemini 2.5 Pro Capable of Winning Gold at IMO 2025*, Huang and Yang [2025], etc.) For example, in Remark 4.5, we show how our TVPAC verifier can be used interactively by a prover that is initially incorrect to arrive at a correct proof in a simplified by rigorous setting. As promised in our rebuttal above, we will emphasize these points in our introduction and take the useful feedback from the reviewer into account.

---

> ### Comment · Reviewer_ANSN · 2025-08-07
> **Thanks for your response.**
>
> **Examples in the Appendix are still abstract.**
> Would it be possible to make a concrete example using Math benchmark problems (like easy problems, GSM8K)?
>
> **Decomposing a CoT thought**
> Can we divide a chunk of text from LLM into separate blocks of tokens?
> Do (Joshi 2025) or (Malach 2024) show how to decompose the text?
> Those papers are more like showing LLM can generate CoT in a theoretical sense, not like providing a way to divide a chunk of text into separate steps. Could you correct if I misunderstood those two papers and your response?
>
>
> **Weak and strong verifiers**
> * weak verifier is a model trained on human annotated CoT dataset (step correctness, the first one is annotated by human).
> * strong verifier: no solution or intermediate evaluation on CoT given. This verifier is a model trained on problem sets with a small fraction of them having a gold standard or one random correct answer.
>
> How to design verifiers from this principle?
> How do you prepare the training set, and what is the sample complexity given the analysis result?
>
>
> **Additional question**
> When we sample a math problem in the whole problem set, some problem instances are similar to each other, but every instance is different. The steps may follow similar mathematical concepts, but each step is all different; at least some numbers plugged into the formula used are different.
> Both weak and strong verifiers make correct decisions on unseen problems in the training set, when some labels are given in that training set. I am confused about this. Maybe GSM8K dataset is an easy case to illustrate the idea.
> Could you explain why such learning is possible?

---

> ### Author Response · Authors · 2025-08-08
> **Thank you for the follow-up**
>
> We thank the reviewer for taking the time to read our response and following up. Please find further clarification below.
>
> **Connection with Math benchmark problems like GSM8K.**
> The GSM8K dataset consists of grade school math problems and was introduced by Cobbe et al. who train verifiers on the reasoning traces generated by the generator (large reasoning model). The difference between our setting and theirs is that we assume we have the correctness information for the reasoning trace up to the first incorrect step, while they use the exact correctness (whether the solution matches tokenwise) to serve as the correctness for every token in the reasoning trace generated by the generator. Here the reasoning steps consist of single calculations expressed as an English sentence, see Cobbe et al. for examples.
>
> **Decomposing a CoT thought.**
> In addition to the GSM8K example above, there has been theoretical work studying LLM generation in terms of reasoning steps (e.g. Shai Shalev-Shwartz and Amnon Shashua [2025]). While Joshi et al. (2025) and Malach (2024) motivate their framework as token-wise generation, their analysis also applies to generating reasoning steps consisting of a collection of tokens by replacing the set $\Sigma$ of tokens by a sequence $\Sigma^{\le n}$ of tokens of length at most $n$ (i.e. larger vocabulary size, but sample complexity bounds are still reasonable).
>
> **Weak and strong verifiers.**
> To design the weak/strong verifiers, one could simply find the verifier in the verifier class (e.g. neural networks with a specific architecture) that minimizes the training error on a training set of sufficient size. As the reviewer notes, one way to prepare the training sets is by human annotation (semi-supervised or weakly supervised extensions are interesting research questions). We provide various sample complexity upper bounds in our paper that depend on the verification goal and the complexity of the verifier class, for example scaling with $\log |H|$ for finite classes $H$ (Theorems 3.2 and 4.3) and with the VC dimension of $H$ for infinite classes (Theorems 3.3 and 4.4). Please refer to the following table for a summary of the data, learning algorithms and sample complexities for the different verification goals.
> |                                 |   SVPAC                    |  TVPAC                 | $\gamma$-TVPAC        |
> | :---------------------  |  ----------------------- | -----------------------| ----------------------- |
> | Data format                     | random (problem, reasoning, first incorrect step) pairs | (problem, $\le k$ gold  standard solutions)| random (problem,  correct reasoning)  pairs     |
> | Learning Algorithm              |  ERM on training set       |ERM using trees $\mathcal{T}_g(x)$|  Intersection of all consistent verifiers (Algorithm 1)         |
> | Sample complexity (finite $H$)  | $\tilde{O}(\log \|H\|)$      |  $\tilde{O}(\log \|H\|)$ | $\tilde{\Theta}(\|H\|)$  |
> | Sample complexity (bounded VC dimension, VCdim($H$))| $\tilde{O}(\text{VCdim}(H))$      | $\tilde{O}(\text{VCdim}(H))$  | $\tilde{O}(\text{VCdim}(H))$ (if intersection-closed) |
>
> **Additional question.**
> It is correct that different problems will have different proofs and the verifier will encounter unseen problems and steps that were never encountered in the training set. The verifier's task is to check stepwise if the last step in the proof follows from all the steps so far. For example, for a step-by-step arithmetic reasoning showing the digitwise calculation for adding two positive integers, the verifier needs to simply be able to check if the carry-on was done correctly and be able to check additions of up to three single-digit numbers.

---

### Note · Authors · 2025-08-15

The authors would like to thank the reviewers and the Area Chair for their time and efforts, and take the opportunity to make the following final remarks.

**On Chain-of-Thought terminology**: We agree with the reviewers that our framework does not involve Chain-of-Thought prompting, but our usage matches the sense in which it was studied in prior theoretical work on sample complexity of Chain-of-Thought generation e.g. Joshi et al., (COLT 2025), Malach, (ICML 2024).

**Tokens vs. reasoning steps**: While next token prediction is the standard approach in LLM generation, for natural language verifiers, it makes more intuitive sense to allow semantic units of reasoning consisting of a few tokens (note that technically our theory also applies to the tokenwise verification case by setting $\Sigma$ appropriately).

**Concrete examples**: Our theory is applicable to designing verifiers for natural language solutions of math problems or logic puzzles. A relevant prior practical work showing usefulness of the verifiers is Cobbe et al. (2021) but they follow a very stringent criterion of exactly matching the gold solutions tokenwise (see previous point).

**Different verification goals**: We have summarized our different verification goals in our rebuttal and responses (e.g. see the Table in response to Reviewer ANSN). They capture different levels of robustness (to distribution shifts in the reasoning traces) and data formats (gold standard proofs, random correct proof etc.), and our sample complexity bounds capture inherent challenges as well as positive results on the amount of data that is sufficient to learn a good verifier.

We thank the reviewers once again for raising several interesting questions.

---

### Decision · Program_Chairs · 2025-09-17

**Decision:**

Accept (poster)

**Comment:**

The paper introduces a PAC‑learning framework for verifiers that assess step‑wise Chain‑of‑Thought (CoT) reasoning, distinguishing simple (SVPAC), trustable (TVPAC) and strong (‑TVPAC) verification goals. It delivers new sample‑complexity upper and lower bounds for finite and VC‑dimension‑bounded verifier classes, clarifies the role of gold‑standard solutions, and connects these theoretical results to concrete settings such as math‑problem benchmarks (e.g., GSM8K) and interactive theorem‑proving scenarios. Reviewers liked the originality of formalizing verification goals, the clarity of key theorems, and the potential impact on reliable LLM reasoning, while noting that the exposition could better situate the work within existing LLM‑verification literature and provide more concrete examples. It seems that the authors’ rebuttal addressed most concerns. Given the solid theoretical contributions, relevance to emerging verification pipelines for LLMs, and the overall consensus that leans to towards accept as well as my own reading, I am recommending acceptance.